



# Alicenet - An Italian network of Automated Lidar-Ceilometers for 4D aerosol monitoring: infrastructure, data processing, and applications

Annachiara Bellini[1,2,3,*], Henri Diémoz[3], Luca Di Liberto[2], Gian Paolo Gobbi[2], Alessandro Bracci[4], Ferdinando Pasqualini[4], Francesca Barnaba[2,*]

[1] University 'La Sapienza', DIET, Rome, Italy

[2] National Research Council - Institute of Atmospheric Science and Climate, CNR-ISAC, Rome, Italy

[3] ARPA Valle d'Aosta, Saint-Christophe, Italy

[4] National Research Council - Institute of Atmospheric Science and Climate, CNR-ISAC, Bologna, Italy

**Correspondence:** Francesca Barnaba (f.barnaba@isac.cnr.it) and Annachiara Bellini (a.bellini@arpa.vda.it)

**Abstract.** The vertically-resolved information on aerosol particles represents a key aspect in many atmospheric studies, ranging from aerosol-climate interactions to those investigating aerosol impacts on air quality and human health. This kind of information can be primarily derived by lidar active remote sensing and extended networks of these systems are currently run at the global scale. A network of Automated Lidar-Ceilometers (ALCs), Alicenet, was set up in Italy in 2015 by the Institute of Atmospheric Sciences and Climate (ISAC) of the National Research Council (CNR). Alicenet grew up in these years as a cooperative effort of Italian institutions dealing with atmospheric science and monitoring, and includes regional Environmental Protection Agencies, Universities and Research Centres. In the current configuration, the network runs both single-channel ALCs and dual channel, polarisation-sensitive systems (PLCs) operating in very different environments (urban, coastal, mountainous and volcanic areas) from Northern to Southern Italy, thus allowing the continuous monitoring of the aerosol vertical distribution across the country. Alicenet also contributes to the EUMETSAT program E-PROFILE, filling an Italian observational gap compared to other EU Member States. In this work, we present the Alicenet infrastructure and a detailed description of the specifically-developed data processing chain converting raw instrumental (Level 0) data into quantitative information on aerosol properties, with output products ranging from attenuated backscatter to aerosol mass and vertical stratification from the surface up to the upper troposphere (output data Levels 1-3). Overall, this setup provides from near real-time to long-term overviews of the 4D aerosol field over Italy. Examples of both are reported in this work. Specific comparisons of the Alicenet products to relevant independent measurements of different parameters (e.g. surface PM10, sunphotometer AOD) are also included, revealing the good performances of the Alicenet algorithms. Overall, Alicenet represents a valuable resource to extend the current aerosol observational capabilities in Italy and in the Central European Mediterranean area, and contributes to bridge a gap between atmospheric science and its application to specific sectors, among which: a) air quality, b) solar energy, c) aviation safety. More in general, the maturity of the ALC/PLC





instrumentation and of the data processing tools available within the wider international scientific/technical atmospheric
community suggest lidar-ceilometer networks could usefully integrate current EU Atmospheric Research Infrastructures for
aerosol studies.

**1. Introduction**

Aerosols influence the Earth radiation budget directly by extinction of solar radiation and indirectly by modification of cloud
properties and lifetime, thus also influencing the hydrological cycle (IPCC, 2022). Atmospheric particles of both
anthropogenic and natural origin are also a main concern for human health worldwide (WHO, 2021). High aerosol loads also
reduce visibility and, particularly during major events (e.g. desert dust storms, volcanic eruptions, wide forest fires), could
represent a threat for the transport sector, and particularly for aviation (e.g. Flentje et al., 2010; Papagiannopoulos et al,
2020, Brenot et al., 2021; Monteiro et al., 2022). A key aspect to correctly quantify aerosol impacts on climate and society is
represented by the aerosol vertical distribution. In fact, it affects radiative transfer balance and atmospheric heating rate (e.g.,
Fasano et al., 2021; Fountoulakis et al. 2022), aerosol-cloud-precipitation interactions (e.g., Napoli et al., 2022), and large-
to-small scale atmospheric processes influencing local air quality (e.g., Curci et al., 2015; Gobbi et al., 2019; Diémoz et al.,
2019a,b) and high-elevation environments (Balestrini et al., 2024).
Active remote sensing through lidar sensors is a very efficient tool to provide range-resolved, accurate profiles of aerosol
properties (e.g., Gobbi et al., 2001; Tesche et al. 2009; Ansmann et al., 2011). In the last decades, both ground-based and
space-based lidar systems have been developed and widely used for scientific research purposes, and they are expected to
play an increasingly important role in climate studies and public health (Remer et al., 2024). From space, the recently
dismissed NASA-CNES CALIPSO sensor (Winker et al., 2010) provided one of the most valuable, vertically-resolved,
global aerosol datasets (2006-2023), that is expected to be extended by the upcoming ESA mission EarthCARE (Cloud,
Aerosol and Radiation Explorer, Illingworth et al., 2015). From the ground, lidar remote sensing is often performed in the
framework of globally distributed research networks. In Europe, a wide Aerosol Research Lidar Network (EARLINET,
Pappalardo et al., 2010) has been developed in the last decade, which is currently an important component of the European
Strategy Forum on Research Infrastructures (ESFRI) ACTRIS. Such a research-oriented network runs high power, multi-
wavelength Raman lidar systems, which were not designed for monitoring purposes. In fact, ACTRIS lidar measurements
are generally not performed continuously, and the spatial density of the measuring sites (about 30 active systems covering
Europe) is still insufficient to capture the high spatial and temporal variability characterising aerosols. These high-power
systems are also affected by a thick blind region in the lowermost atmospheric levels, which strongly limits their utility in
monitoring boundary layer processes. Only recently, ACTRIS started considering using automatic low-power lidars
(hereafter referred to as Automated Lidar-Ceilometers, ALCs) as useful tools within its aerosol remote sensing (ARS)
component, although these systems are not yet included in the relevant 'minimum' or 'optimal' setups recommended by
ACTRIS-ARS (https://www.actris.eu/topical-centre/cars/announcements-resources/documents, last access 7 March 2024),
and are rather only included within the ACTRIS-Cloud remote sensing component for liquid-cloud detection.



In fact, in the last two decades, technological improvements allowed the development of low-energy, eye safe, affordable
and robust single-channel ALCs, originally conceived to only monitor the 'cloud ceiling'. These systems emit single-
wavelength laser pulses, mostly in the infrared range and, similarly to high power lidars, measure the time- (thus range-)
dependent radiation elastically scattered back to the instrument by atmospheric components (molecules, aerosols, cloud
droplets/ice crystals). Although with a lower signal-to-noise (SNR) ratio with respect to their high-power counterpart, a
major advantage of ALCs is their ability to operate continuously, reliably and unattended, providing continuous information
on aerosol profiles and clouds within the troposphere in near-real time. This favoured the development of extended ALC
networks worldwide. These include the NASA Micro-Pulse Lidar Network (MPLnet; Welton et al., 2018), the US
Environmental Protection Agency (EPA) network for Photochemical Assessment Monitoring Stations (PAMS; Caicedo et
al., 2020), or the Asian Dust and aerosol lidar observation network (ADnet; Shimizu et al., 2016). In Europe, several
Member States currently run dense ALC networks for monitoring purposes, mostly managed by National Meteo Services
(e.g., the DWD in Germany or the MetOffice in the UK; Flentje et al., 2021; Osborne et al., 2022). In Europe, most of such
ALC observations are also collected and further exploited in the framework of the E-PROFILE program run by the European
Meteorological Services Network EUMETNET ([http://www.eumetnet.eu/activities/observations-programme/current-](http://www.eumetnet.eu/activities/observations-programme/current-activities/e-profile/)
[activities/e-profile/](http://www.eumetnet.eu/activities/observations-programme/current-activities/e-profile/), last access: 06-03-2024). The development of such an extended ALC observational capacity was very
useful, for example, during the eruption of the Icelandic volcano Eyjafjallajökull in 2010, during which a readily accessible
information on the aerosol plume horizontal and vertical displacement was needed, considering the major impact on aerial
transport (Flentje et al., 2010, Mortier et al., 2013). More in general, ALC data can be used to study a variety of atmospheric
processes related to aerosols, fog (Haeffelin et al., 2016), and clouds (Van Tricht et al., 2014), with benefits for a range of
users in several socio-economic sectors. In particular, ALCs have been proven to be extremely useful in support of surface
Air Quality (AQ) evaluations. In fact, they can provide information on the vertical dilution of pollutants, and are able to
detect transboundary transport of particles from medium-to-long-range distances (e.g., Bucci et al., 2018; Diémoz et al.,
2019a,b), secondary aerosol layers forming aloft, or even particles  reaching down to the boundary layer through evaporating
rain (virgas, e.g., Karle et al., 2023). However, with few exceptions, standard Air Quality Monitoring Networks (AQMNs)
currently miss such profiling capability. The feasibility of filling this gap is currently explored in the framework of the EC-
H2020 Project RI-URBANS ([https://riurbans.eu](https://riurbans.eu), last access: 7 March 2024), aiming at the development of service tools in
support to urban AQ monitoring in European urban centres and pollution hotspots. In fact, the current ALC technology has
been proven to be mature enough to allow a robust retrieval of the planetary boundary layer height (Kotthaus et al., 2023), a
key parameter in AQ, and evaluations are currently ongoing at the EU level to assess readiness of ALC-based retrievals for
quantitative Particulate Matter (PM) monitoring (e.g., Shang et al., 2021). This is done for example in the framework of the
EU Action PROBE (PROfiling the atmospheric Boundary layer at European scale; Cimini et al., 2020) supported by the
European Cooperation in Science and Technology (COST), as well as within the E-PROFILE and ACTRIS communities.
In Italy, an effort to coordinate ALCs activity at national level has been done by the National Research Council -Institute of
Atmospheric Sciences and Climate of the (CNR-ISAC), which set up the Alicenet network in 2015 ([https://www.alice-](https://www.alice-)



net.eu/, last access: 7 March 2024) and is currently contributing to E-PROFILE, filling an observational gap over Italy, if
compared to other EU Member States. These observations are particularly relevant for the Mediterranean area, this being a
climatic hotspot (IPCC, 2022) and a crossroad of large scale fluxes from Continental Europe, North America, Africa and
Asia (Lelieveld et al., 2002). In this area, the atmospheric circulation also features a wide spectrum of mesoscale fluxes
triggered by complex topography. The resulting aerosol mixture is also a complex cocktail of anthropogenic, marine, desert
dust, and biomass burning particles (e.g., Barnaba and Gobbi, 2004; Di Iorio et al., 2009, Andres Hernandez et al., 2022).
The present work aims at presenting the Alicenet infrastructure and its data processing chain, specifically designed to derive
quantitative information on vertically-resolved aerosol properties and layering. Alicenet is an open consortium coordinated
by CNR-ISAC with contributions from several collaborating Partners (regional Environmental Protection Agencies,
Universities, Research Institutions, and private companies). The ALC data processing is centralised at CNR-ISAC, allowing
the retrieval of homogeneous quantitative ALC-based aerosol information from North to South Italy. The Alicenet
processing chain is based on specifically developed retrieval algorithms, taking benefit from past and ongoing collaborations
with the EU ALC-community, and particularly in the framework of the EU COST Actions TOPROF (2013-2016) and
PROBE (2019-2024), the H2020 Project RI-URBANS, and the E-PROFILE initiative.
The work is organised as follows: Sect 2 describes the Alicenet infrastructure. Sect. 3 introduces the data processing and
includes specific examples of the Alicenet-based products further compared to independent datasets. In Sect. 4, three recent
examples of the near-real time Alicenet monitoring capability  are reported. A final summary is given in Sect. 5, this also
describing some foreseen future developments.

## 2. Alicenet sites and instruments

The Alicenet stations are geographically distributed from the North to the South of the italian peninsula as shown in Fig. 1.
The network configuration also allows the monitoring of aerosol vertical profiles over a wide range of environmental and
atmospheric conditions (e.g. urban, coastal and mountain) within the Mediterranean area. In fact, some sites are located in
highly anthropized areas, such as those in the Po Valley and main urban/industrial sites in Italy (Milan, Genova, Turin,
Rome, Taranto). Most sites are frequently impacted by desert dust advections, particularly relevant in Central and Southern
Italy (e.g. Barnaba et al., 2017; Gobbi et al., 2019; Barnaba et al., 2022), and vegetation fires (e.g. Barnaba et al., 2011).
Episodes of volcanic plumes transport are also registered in the Alicenet southernmost sites, mostly in the 5 stations located
at the foothills of the Etna volcano, and in the Messina and Lamezia Terme stations, due to their proximity to the other active
sicilian volcano of Stromboli.





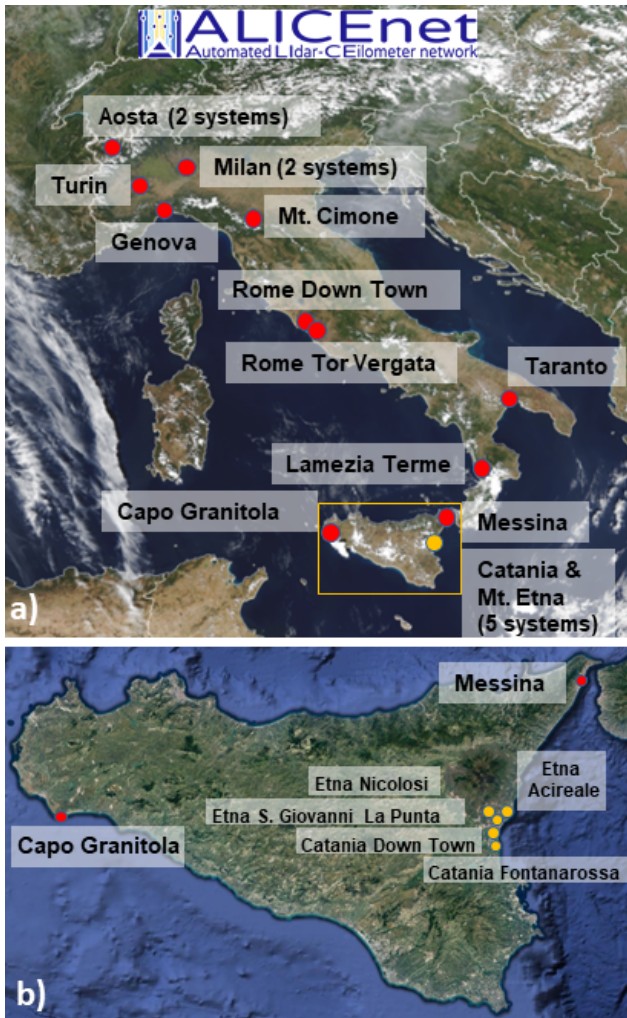

**Figure 1:** Location of the Alicenet stations (panel a, bullets). The yellow rectangle over Sicily in panel a) is zoomed in panel b) to show location of the 5 stations in the area extending from the Etna volcano southern foothills to the city of Catania. Maps: a) satellite true colour image (credits: EUMETSAT), and b) credits: © Google Maps.

For homogeneity of operations, since the beginning of the Alicenet activities (1ˢᵗ January 2016), it was agreed to operate standardised systems across the network choosing the ones having a sufficiently high SNR. The single-channel CHM15k instruments manufactured by Lufft (formerly Ott Hydromet) were selected for this purpose. These are bi-static ALCs with a Nd : YAG solid-state laser emitting linearly polarised light at 1064 nm, with a 5-7 kHz repetition rate, a maximum vertical resolution of 5 m and a maximum range of 15 km. The only exception in this instrumental setup was a modified-CHM15K prototype with polarisation-sensitive capabilities developed in 2013 by Jenoptik ESW in the framework of the EC Life+





DIAPASON project (Gobbi et al., 2019). This first ever polarisation-sensitive ALC (hereafter PLC) was intended to explore
the possibility to produce an affordable, robust system to be widely used in the identification and profiling of non-spherical
(e.g. mineral dust) aerosol layers. The prototype PLC was tested in Rome (Italy), where it has been operating successfully
since then (e.g., Gobbi et al., 2019; Andres Hernandez et al., 2022), but was never marketed by Lufft. More recently, PLC
systems have been made available on the market by Vaisala (CL61 systems, operating at 910 nm) and, due to the important
capability of such instruments to discriminate particle sphericity/non sphericity, these are being progressively included in
Alicenet.
For both CHM15k ALCs and CL61 PLCs, the signal is characterised by high temporal and vertical resolution, with some
variability depending on the system type and configuration (e.g., in Alicenet the CHM15k standard configuration implies a
vertical (temporal) resolution of 15 m (15 s)). A summary table with details on the Alicenet sites and instrumentation
operating therein is provided in Table 1.

| Name | Lat | Lon | Altitude (m asl) | System Type | Reference Institution (& Collaborating Institution) |
|---|---|---|---|---|---|
| Aosta 1 | 45° 44' 32" N | 07° 21' 24" E | 555 | ALC (Lufft CHM15K) | Arpa Valle d'Aosta (CNR-ISAC) |
| Aosta 2 | 45° 44' 32" N | 07° 21' 24" E | 555 | PLC (Vaisala CL61) | Arpa Valle d'Aosta (CNR-ISAC) |
| Milano Bicocca | 45° 30' 38" N | 09° 12' 42" E | 135 | ALC (Lufft CHM15K) | CNR-ISAC (Univ. Milano Bicocca) |
| Milano Rubattino | 45° 28' 38" N | 09° 15' 41" E | 110 | PLC (Vaisala CL61) | RSE (CNR-ISAC) |
| Torino | 45° 03' 28" N | 7° 39' 24" E | 250 | PLC (Vaisala CL61) | Politecnico Torino (CNR-ISAC) |
| Genova | 44° 24' 41" N | 08° 53' 3" E | 10 | PLC (Vaisala CL61) | Arpa Liguria (CNR-ISAC) |
| Monte Cimone | 44° 11' 35" N | 10° 42' 05" E | 2165 | ALC (Lufft CHM15K) | CNR-ISAC |



| | | | | | |
|---|---|---|---|---|---|
| Rome Down Town | 41° 54' 34" N | 12° 29' 48" E | 58 | PLC (Lufft Prototype) | CNR-ISAC (Arpa Lazio) |
| Rome Tor Vergata | 41° 50' 32" N | 12° 38' 50" E | 100 | ALC (Lufft CHM15K) | CNR-ISAC |
| Taranto | 40° 29' 37" N | 17° 13' 01" E | 17 | ALC (Lufft CHM15K) | Arpa Puglia (CNR-ISAC) |
| Lamezia Terme | 38° 52' 35" N | 16° 13' 56" | 5 | ALC (Lufft CHM15K) | CNR-ISAC |
| Messina | 38° 11' 41" N | 15° 34' 22" E | 5 | ALC (Lufft CHM15K) | CNR-ISAC (CNR-IRBIM) |
| Etna Acireale | 37° 38' 26" N | 15° 10' 55" E | 12 | ALC (Lufft CHM15K) | Etna High Tech (CNR-ISAC, INGV) |
| Etna Nicolosi | 37° 36' 49" N | 15° 01' 11" E | 730 | PLC (Vaisala CL61) | INGV (CNR-ISAC, Etna High Tech) |
| Etna San Giovanni La Punta | 37° 34' 44" N | 15° 06' 11" E | 350 | ALC (Lufft CHM15K) | Etna High Tech (CNR-ISAC, INGV) |
| Capo Granitola | 37° 34' 16" N | 12° 39' 35" E | 5 | ALC (Lufft CHM15K) | CNR-ISAC |
| Catania Down Town | 37° 30' 49" N | 15° 04' 55" E | 40 | ALC (Lufft CHM15K) | INGV (CNR-ISAC, Etna High Tech) |
| Catania Fontanarossa | 37° 27' 59" N | 15° 04' 57" E | 10 | ALC (Lufft CHM15K) | SAC (Etna High Tech, CNR-ISAC, INGV) |

**Table 1:** Alicenet sites from northern to southern Italy, and relevant details.





## 3. Alicenet data processing and relevant products

As in any backscatter lidar, the raw signal recorded by the ALCs is a function of the distance from the emitter (range, r) and the emission/reception time t, and can be described through the lidar equation:

$$P(r,t)=r^{-2}Ovl(r,t)C_L(t)\left(\beta_p(r,t)+\beta_m(r,t)\right)e^{-2\int_0^r\left(\alpha_p(r',t)+\alpha_m(r',t)\right)dr'} \qquad (1)$$

Equation 1 includes the particles (p) and molecules (m) backscatter ($\beta$) and extinction ($\alpha$) coefficients at the laser wavelength (elastic scattering), and some instrumental factors, embedded into the instrument-specific calibration coefficient $C_L(t)$. Furthermore, particularly for bistatic systems (i.e., the CHM15k), measurements in the near range are generally affected by signal losses due to the incomplete superposition (overlap) of the laser beam and the receiver field of view, this depending on the range and the time-varying instrument characteristics. The term Ovl(r,t) in Eq. 1 therefore indicates an instrument-specific overlap function used to quantify the signal loss in the near range. The total attenuated backscatter, $\beta_{att}$, i.e., the total (aerosol + molecules) backscatter coefficients attenuated by the total (aerosol + molecular) atmospheric transmission, can be simply derived from Eq. 1 as:

$$\beta_{att}(r,t)=\frac{P(r,t)r^2}{Ovl(r,t)C_L(t)}=\left(\beta_p(r,t)+\beta_m(r,t)\right)e^{-2\int_0^r\left(\alpha_p(r',t)+\alpha_m(r',t)\right)dr'} \qquad (2)$$

In this section, the Alicenet data processing chain and main inputs and outputs (Fig. 2) are described. Each step of the processing chain is functional to derive specific terms of Eq. 1, and finally retrieve quantitative aerosol information from the ALC raw data. The procedure takes as input the ALC data formatted through the E-PROFILE Raw2L1 routine (main variables are the range-corrected signal, RCS(r, t) = P(r, t) $r^2$, and the volume linear depolarisation ratio, D(r,t)).

The whole Alicenet data processing chain is developed for CHM15k systems since, as mentioned above, these were the ones firstly implemented in the network. It includes pre-processing procedures (cloud screening, denoising, overlap correction; Sect. 3.1), the absolute calibration (to determine $C_L$ and, in turn, $\beta_{att}$; Sect. 3.2), the quantitative retrieval of aerosol optical ($\beta_p$ and $\alpha_p$) and physical (surface area, $S_p$, volume, $V_p$, and mass concentration, $M_p$) properties (Sect. 3.3), and the automatic identification of main aerosol layers (continuous and mixed, CAL and MAL, and elevated, EALs, aerosol layers; Sect. 3.4). A similar full inversion chain is under development for CL61 systems, for which data processing is currently limited to the cloud screening and denoising, the absolute calibration (to monitor the stability of the instrument calibrated by the manufacturer), and the identification of aerosol layers (Fig. 2). Exploiting the CL61 depolarisation measurements (e.g., Tesche et al., 2009), for these systems, the aerosol type dominating within each detected layer is also estimated (see Sect. 3.4). For clarity, in Fig. 2 different colours are used to indicate Alicenet inputs, inversion steps, and outputs valid for CHM15k only (light green), for CL61 only (cyan), or for both (dark green).



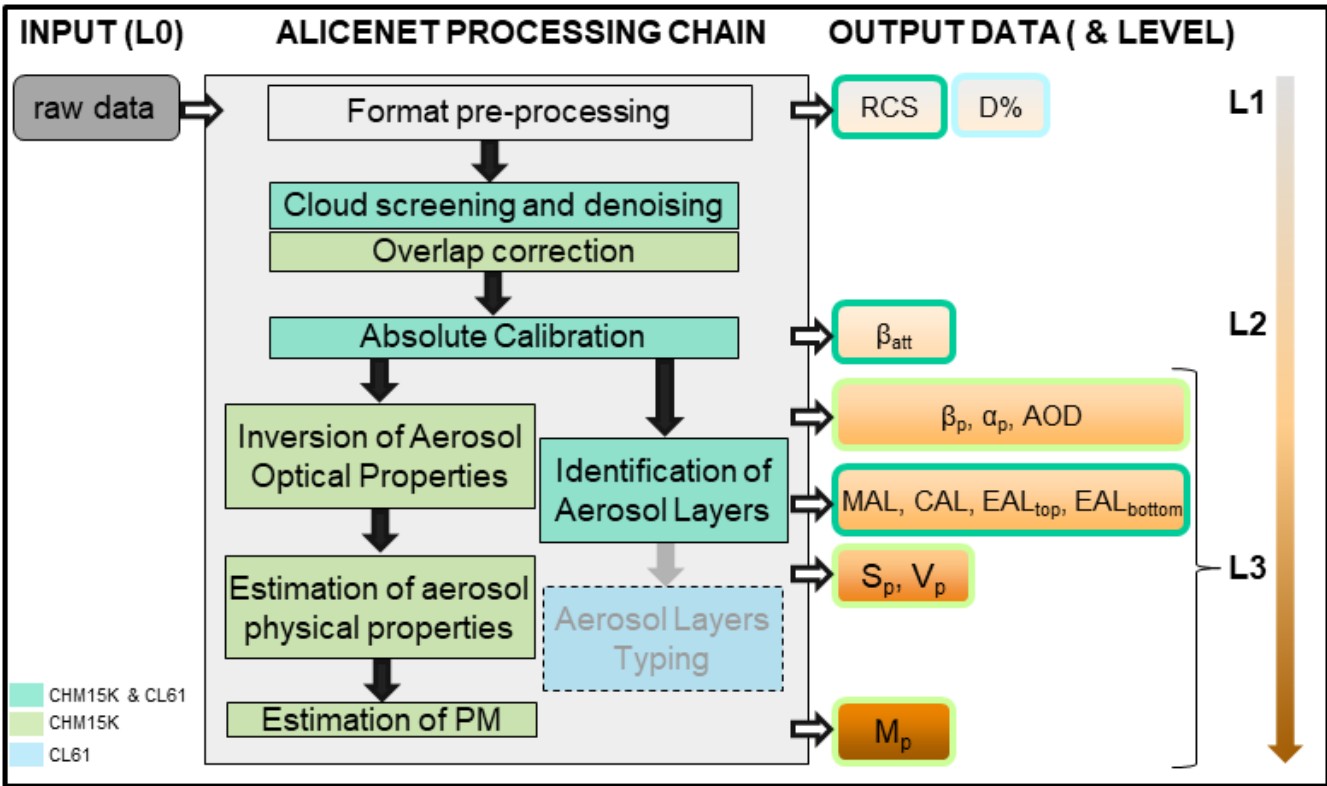

**Figure 2:** Scheme of the Alicenet processing chain of the raw (L0) data and relevant output data products (L1-L3). The different colours in the processing box are used to indicate inversion steps valid for CHM15k (light green), CL61 (cyan), or both (green) system types. This same colour code is used for relevant output data products, which are further coloured from light to dark orange indicating processing level, from the more basic (L1) quantities (altitude dependent Range-Corrected Signal, RCS, and depolarisation, D), through the attenuated backscatter ßatt (L2) to the optical and physical aerosol properties and aerosol layering (L2 & L3), see text for further details.

The Alicenet processing chain is completely automatic and centralised at CNR-ISAC. This setup allows the continuous monitoring of the aerosol distribution across Italy, with L1 and/or L2 information accessible in near-real time through a dedicated website (https://www.alice-net.eu/, last access: 7 March 2024). Some selected examples of this monitoring capability are provided in Sect. 4. The more advanced, homogeneous and quantitative retrieval of aerosol properties and layering across the network (L3 products) is conversely performed in post-processing and is not yet provided through the Alicenet webpage.

**3.1 Pre-processing**

The first pre-processing steps are aimed at a) avoiding cloud/precipitation contamination in aerosol retrievals, and b) tackling the signal noise (Sec. 3.1.1). Then the data need to be corrected for overlap artefacts (Sec. 3.1.2) before proceeding



with the determination of the instrument-specific calibration coefficient (Sec. 3.2). The way these preliminary steps are
performed within Alicenet is described hereafter.

### 3.1.1 Cloud-screening and denoising

At the ALC laser wavelengths (near infrared), the reflectivity and optical thickness of clouds produce complete extinction of
the laser beam above the cloud base. Only in case of thin clouds is the laser beam partially transmitted above the cloud base,
but the return signal has a too low SNR to be employed for aerosol retrievals. The cloud-screening applied to the Alicenet
data exploits the Cloud Base Height (CBH) identified by the ALC firmware (e.g., for CHM15k refer to
https://www.lufft.com/, last access: 7 March 2024). In Alicenet, all data above an altitude of CBH - 500 m are filtered out,
thus avoiding both the presence of cloud droplets and possible virga effects frequently observed below the cloud base. This
also limits  the impact of some variability in the CBH identification depending on the instrument type and firmware version.
A temporal cloud-filter is also further applied, this removing profiles collected 15 min before-to-15 min after the firmware
cloud detection. Some examples of the cloud screening applied to Alicenet ALC signals are reported in the Supplement
(Sect. S1).

After the cloud-screening (if needed), the profiles are downscaled and denoised to improve accuracy of the aerosol retrievals.
Indeed, as mentioned above, the ALC signal is generally collected with high temporal and vertical resolution and is affected
by a decrease of the SNR along the profile. The denoising is performed by computing  signal mean and standard deviation
over specific time and range windows, and filtering those data where the SNR (defined as the ratio between the mean and the
standard deviation) is below a given threshold. A minimum SNR of 20% is generally set for aerosol retrievals within
Alicenet, while the resolution of the downscaled data is modulated depending on the time scales of the processes to be
investigated. It may range from 1 min for the investigation of boundary layer dynamics (as done for the identification of the
mixed aerosol layer) up to 30 min or 3 hours for the identification of aerosol loaded/free regions in the upper troposphere (as
done for the identification of elevated aerosol layers or within the absolute calibration procedure, respectively).

### 3.1.2 Overlap correction

For CHM15K systems, an overlap correction of the signal in the near range (0-1000 m a.g.l.) is required. This is particularly
important when ALC data are used for surface AQ applications, and especially in those conditions in which particulate
matter is confined in the lowermost atmospheric levels. An instrument-dependent, time-invariant overlap function
accounting for the signal loss is generally provided by the manufacturer ($Ovl_{man}(r)$). However, it has been demonstrated
(Hervo et al.,  2016) that changes in the instrument sensitivity rather require the use of a temperature-varying overlap
function. Within Alicenet, the derivation of such an overlap function is based on the procedure developed by Hervo et al.
(2016) to which the reader is referred to for full details. This procedure firstly selects time windows in which it can be
reasonably assumed to have an homogeneous aerosol layer down to the surface. Then, for those selected time windows, the



homogeneity of the ALC signal is forced down to the ground so to derive the overlap correction factor ($f_c(r)$) to be applied to
the manufacturer overlap function. As found by Hervo et al. (2016), $f_c(r)$ is actually dependent on the system internal
temperature. Therefore, to take this dependence into account, an ensemble of overlap correction factors is derived using a
reasonably long dataset (generally spanning different months), and each $f_c(r)$ is put in relation (linear fit) with the median
internal temperature of the system ($T_m$) within the corresponding time interval. The result is a range- and temperature-
dependent 'overlap model' Ovl(r,T). Examples of such overlap models obtained for some Alicenet sites are given in the
Supplement (Sect. S2).
Within Alicenet, some additional requirements and quality controls to the Hervo et al. (2016) procedure were introduced to
derive a robust Ovl(r,T):

 a) since the assumption of aerosol homogeneity in the lowermost levels is rarely fulfilled at some stations, additional

quality controls were introduced on the ensemble of $f_c(r)$ in order to reject those likely derived in inhomogeneous
conditions and thus leading to unrealistic overlap corrections (details are provided in the Supplement, Sect. S2);
b) the instrument-dependent overlap models are derived considering at least a 1-year datasets to reach a statistically
significant ensemble of overlap functions spanning a representative range of temperature;
c) the overlap correction is only applied above 225 m a.g.l., to avoid altitude ranges where the partial overlap is still
insufficient to derive quantitative information. Below this altitude, the profiles are extrapolated down to the ground
either by linear fitting (applied for winter data, using the first four corrected ranges > 225 m) or assuming an
homogeneous profile (applied for summer data).
As an example of the effects of overlap corrections, in Fig. 3 the 24h $\beta_{att}$ profiles derived using (a) the overlap correction
provided by the manufacturer and (b) the Alicenet overlap correction are shown (site: Rome-Tor Vergata, date: 12 August
2019). This case was selected because of the high diurnal variation (15 K) of the system internal temperature. It is evident
that the temperature-dependent overlap model is effective in correcting the false-gradient and the aerosol overestimation in
the lowermost 500 m that the manufacturer function was not able to remove.





**Figure 3:** Overlap-corrected ALC profiles using: (a) the manufacturer overlap correction, and (b) the Alicenet overlap correction (data referring to Alicenet Rome-Tor Vergata site on 12/08/2019).

A further effort to evaluate the ability of this procedure to provide a reliable overlap correction was conducted in the mountain site of Aosta, exploiting the clean, nearly-molecular conditions often registered at this pre-alpine station. In fact, due to its location, Aosta is frequently characterised by relatively low aerosol concentrations in the lowermost levels, these conditions being mainly registered during Föhn events. This makes it possible to compare the overlap-corrected $\beta_{att}$ profiles with a theoretical molecular profile at very low altitudes. To perform this exercise, Föhn-related aerosol-free conditions of 3- to-6 hours were identified by exploiting ancillary aerosol data (namely, surface PM10 concentrations measured by an OPC, sun photometer-derived Aerosol Optical Depth, AOD) and meteorological parameters (wind, pressure, RH) from the AQMN of ARPA Valle d'Aosta (Diémoz et al., 2021). For each of these selected cases, the mean $\beta_{att}$ profiles retrieved using both the



manufacturer overlap correction and the Alicenet overlap correction were compared with a theoretical molecular profile.
Figure 4 shows results for two cases (referring to 25 May 2021 and 6 October 2021) characterised by different values of the
instrument internal temperature (308 K and 292 K, respectively) and low aerosol loads both at the surface ($PM_{10} < 6$ and 5
µg m$^{-3}$, respectively) and along the column (AOD at 1200 nm < 0.04 and 0.03, respectively).

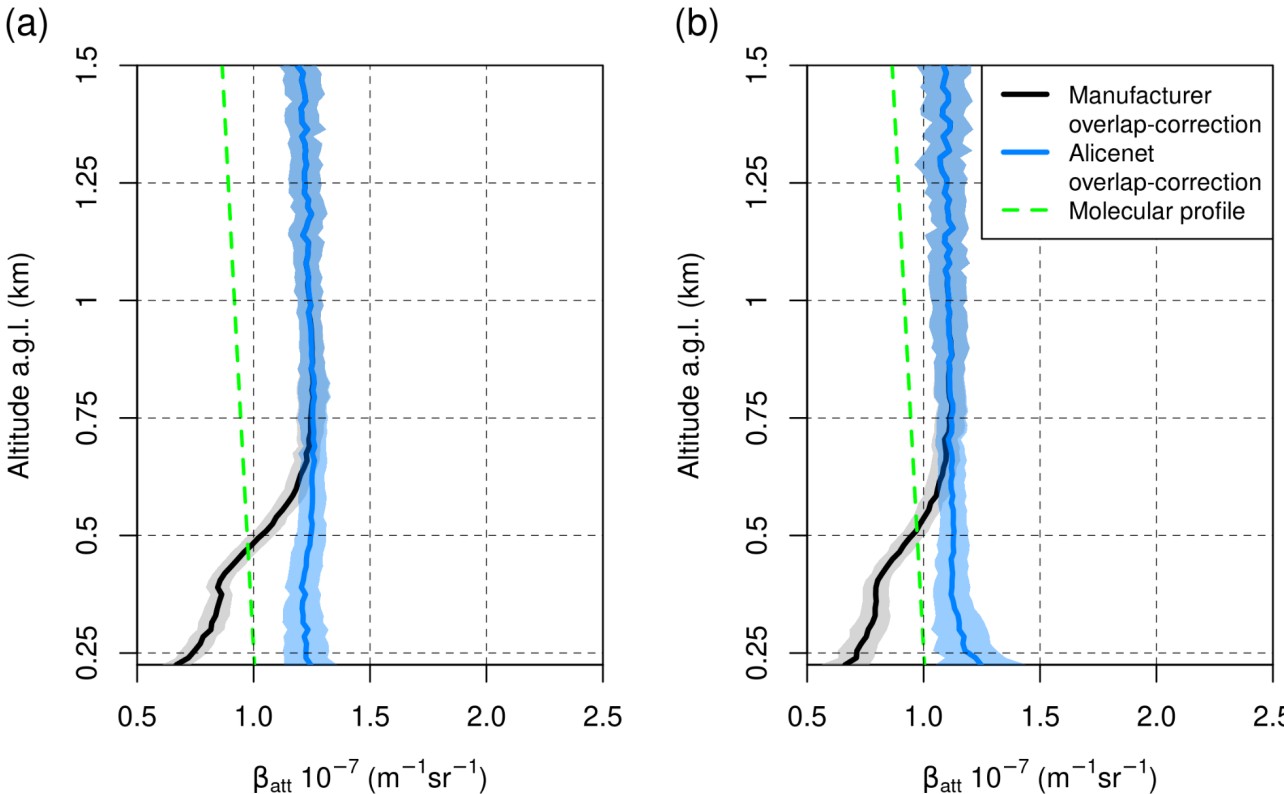

**Figure 4:** $\beta_{att}$ profiles derived using the manufacturer overlap correction (black line) and the Alicenet overlap correction (blue line) in two
nearly-molecular conditions registered in Aosta on: (a) 25 May 2021 (5-8 UTC), and (b) 6 October 2021 (9-12 UTC). The shaded areas
represent the $\beta_{att}$ standard deviations within the selected time windows. The theoretical, molecules-only $\beta_{att}$ profile is also reported (green
line).





Overall, the results show that the Alicenet overlap correction for the CHM15k in Aosta allows to reproduce the nearly-
homogeneous, nearly-molecular $\beta_{att}$ profiles expected in the lowermost levels (< 1500 m), and to avoid unphysical values
(lower than the molecular profile).

**3.2 Absolute calibration**

The absolute calibration (i.e., derivation of the calibration coefficient $C_L$ in Eq. 1) is required to convert the pre-processed
signal into quantitative aerosol information. For CHM15k systems, generally characterised by a SNR > 20% up to 6-7 km,
the absolute calibration procedure is performed by comparing the ALC signal with a theoretical molecular profile (Rayleigh
calibration; Klett, 1985) in aerosol-free atmospheric regions (generally in the mid-upper troposphere). The theoretical
molecular backscatter profile at the operating wavelength is derived based on temperature and pressure data (Bodhaine,
1999). Within Alicenet we use site-dependent monthly averaged profiles of these variables, extracted from ERA5 reanalyses.
This calibration procedure, which is fully automatic, takes as input the pre-processed ALC signals and is made in two steps:
a) search for the best-suitable molecular window, and b) computation of the calibration coefficient. The procedure builds on
the E-PROFILE algorithm ([http://www.eumetnet.eu/activities/observations-programme/current-activities/e-profile/](http://www.eumetnet.eu/activities/observations-programme/current-activities/e-profile/), last
access: 06-03-2024), although Alicenet introduced some specificities and quality controls in both steps as detailed below.
a) The selection of the molecular window is performed during nighttime (this avoiding sunlight noise) using vertical profiles
collected over 3-6 hours (depending on cloudiness) at altitude ranges of 3-7 km altitude. Along this vertical range, an
iterative procedure considers a fine grid of 'potential' molecular windows centred at different altitudes and with variable
amplitudes (from 600 to 3000 m, at steps of 30 m). For each potential range-window (i.e. combination of central altitude and
amplitude), the linear fit between the time-window-averaged signal and the theoretical molecular attenuated backscatter
profile is performed. In order to reject the range-windows affected by aerosol layers, a test is performed to check for the
presence of coherent structures therein. More specifically, the Breusch-Godfrey test (BG test; Breusch, 1978) is applied to
calculate the autocorrelation in residuals, and the windows associated with high autocorrelation (i.e., p-value of the BG test >
0.05) are rejected. The molecular window selected from the ensemble of retained windows is the one maximising a specific
metric which takes into account both the adjusted $R^2$ and the intercept of the linear fit (details are reported in the Supplement,
Sect. S3). The following two quality controls are then performed:
-    The first quality control (QC1) assures that the linear fit slope is positive and the intercept nearly 0;
-    There might be cases in which an undetected homogeneous aerosol layer within the range-window leads to a
303          misleading robust linear regression with the molecular profile. A second quality control (QC2) is thus applied to
304          filter range windows indicating the presence of an aerosol layer within their boundaries (details are reported in the
305          Supplement, Sect. S3).
If these are not met, the night is rejected for calibration purposes, and the process continues using data from the following
day.



b) Once the  quality controlled molecular window is selected from the procedure above, the backward Klett inversion is used
for the inversion of the time-averaged ALC signal to obtain the $\beta_{att}$ profiles following Wiegner and Geiβ (2012; 2014). Note
that the sign correction in the Klett algorithm reported by Speidel and Vogelmann (2023) was already introduced in the
Alicenet procedure. The $C_L$ is then derived as:
$$C_L = \frac{\overline{P(r)\frac{r^2}{Ovl}(r)}}{\overline{\beta_{att}(r)}}$$
(3)

Where the overbar denotes the median along the identified aerosol-free (molecular) range-window.
Two further quality controls are then performed at this stage:
-    A third quality control (QC3) rejects the calibration coefficients for which the uncertainty exceeds a given threshold

316          (see the definition of the $C_L$ uncertainty and the threshold value in the Supplemen (Sect. S3);

-    Since the derived aerosol extinction coefficient can assume slightly negative values if the selected molecular

318          window contains a non-negligible amount of aerosols, a last quality control (QC4) filters those calibrations

319          associated with a negative sign of the AOD in the atmospheric layer 0-4 km resulting from the Klett inversion.

Figure 5 (top panels) shows two examples of successful calibrations, referring to the Alicenet ALC system operating in
Aosta, during the nights of 21 May and 25 October 2017, respectively. The selected spring (autumn) nighttime calibrations
correspond to a $C_L$ close to the maximum (minimum) value of the year 2017. This can be verified in Fig 5c, showing the 7-
year $C_L$ time series (2016-2022) derived for the Alicenet CHM15k systems operating in Aosta, Rome and Messina (i.e.,
Northern, Central and Southern Italy, respectively).
It is evident from Fig. 5c that the three series feature a similar seasonal cycle, as also observed in other European sites
(https://www.probe-cost.eu/resources/, last access: 7 March 2024). The reason for such a cycle is not clear yet, and is
currently under investigation. Main hypotheses are this cycle being related to instrumental issues (e.g. temperature-related)
and/or to atmosphere-related phenomena. Different initiatives within the EC Cost Action PROBE, in which Alicenet is also
involved, are currently ongoing to shed light on this seasonality. As reported by PROBE released documents, an undetected
500nm-AOD of 0.02 within the calibration window would be sufficient to perturb the calibration coefficient up to 30%, i.e.,
the maximum amplitude of the seasonal cycle observed in Aosta and Messina. Such an increase of the AOD within the
calibration window is indeed plausible in spring and summer due to convection, large scale transport processes, and
photochemical reactions.



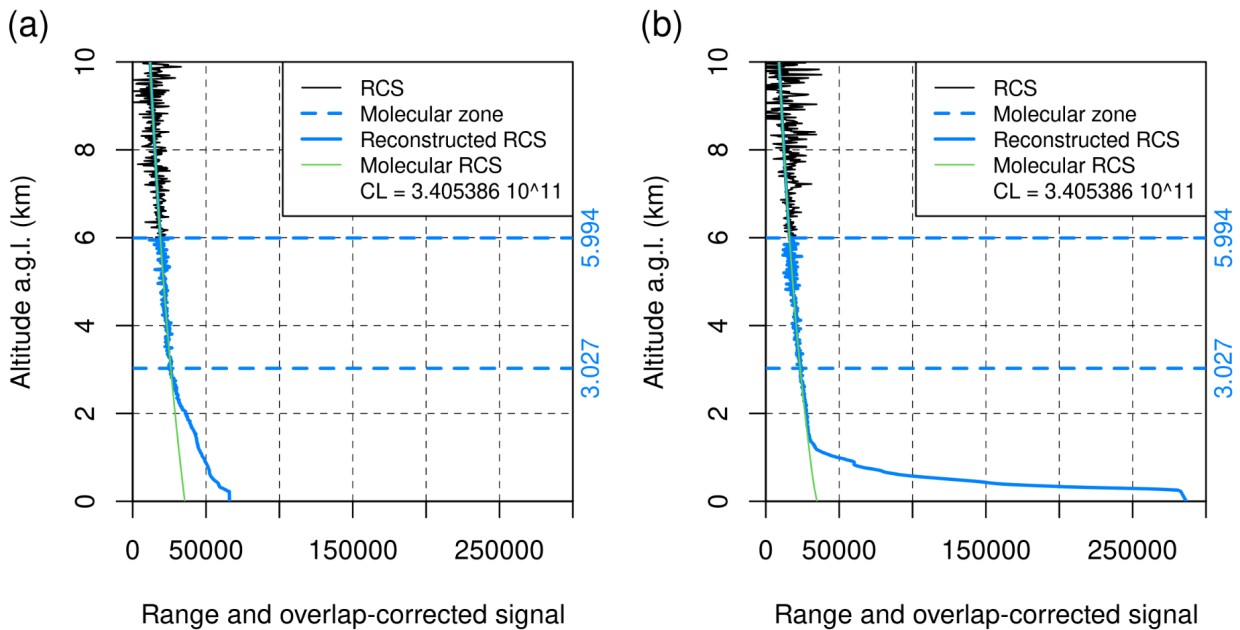

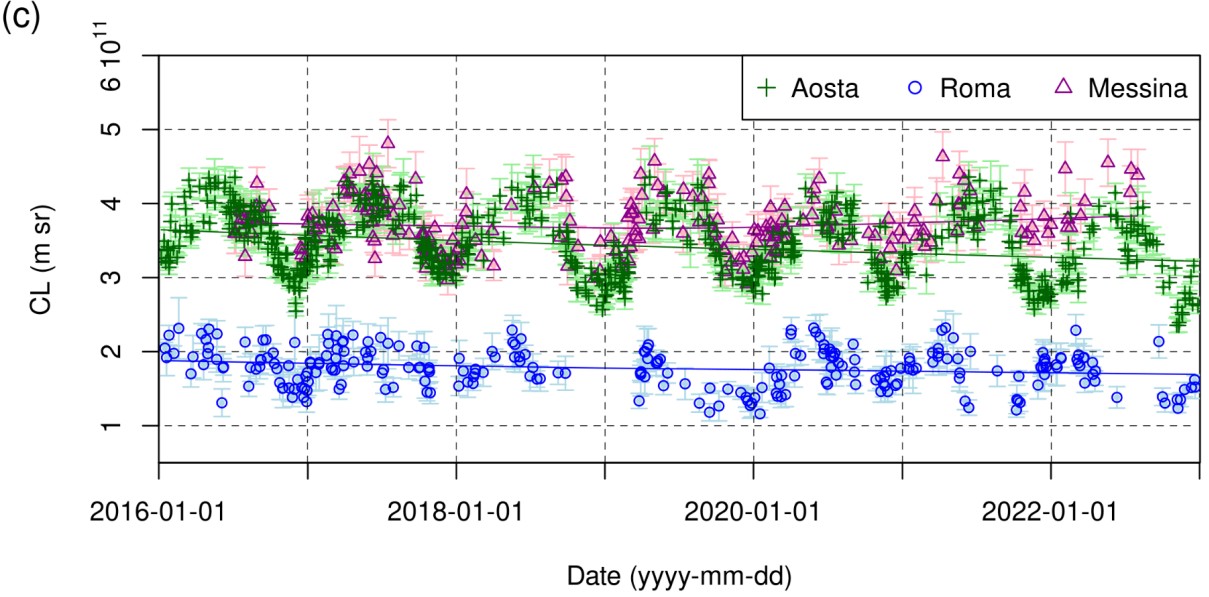

**Figure 5:** (a, b) Examples of output of the Alicenet calibration procedure (nighttime signals of the Aosta CHM15k on 21 May and 25 October 2017, respectively) with indication of the molecular zones selected, and derived CL values. (c) Long-term (2016-2022) time series





of the calibration coefficients ($C_L$ ) derived for the  CHM15k systems operating in Aosta, Rome, and Messina, and associated Loess fits
(lines) used to derive the actual CL values used in the operational, all-year-round data-inversions.

Due to the unclear reason for the variability of the calibration coefficients, at present the $C_L$ values operationally used within
Alicenet for the site-and day-specific aerosol retrievals are derived through a locally weighted smoothing (Loess) fit with a
time span > 1 year, thus flattening the seasonal variability but following long term trends related to instrument ageing (Fig.
5c). It is important to underline that a different treatment of this variability could be necessary once the main driver of the $C_L$
seasonality is better identified. The uncertainty in the aerosol retrievals resulting from the $C_L$ variability is discussed in Sect.

346 3.3.3.

**3.3 Retrieval of aerosol properties**

This section describes the Alicenet inversion of the aerosol optical (Sect. 3.3.1) and physical (Sect. 3.3.2)  properties.
Specific examples of retrievals at the different Alicenet sites are also given and compared to a series of independent datasets
in order to evaluate the relevant retrieval procedure performances.

**3.3.1 Aerosol optical properties**

The aerosol backscatter ($\beta_p$) profile is calculated from the total attenuated backscatter ($\beta_{att}$) profile based on the forward Klett
inversion (Wiegner and Geiβ, 2012; 2014) of Eq. 1. Since both aerosol extinction and backscatter are unknown in Eq. 1, an
assumption on the relationship linking the two variables is necessary to solve the Klett inversion. Within Alicenet, we do not
fix an a priori, vertically-constant extinction-to-backscatter ratio (also referred to as  lidar ratio, LR) , as often done in elastic
lidar retrievals. Instead, the aerosol extinction is assumed to be linked to backscatter through a specific functional
relationship ($\alpha_p=\alpha_p(\beta_p)$). This was obtained at the CHM15K operating wavelength (1064 nm) using a continental-type aerosol
model as described in Dionisi et al. (2018). More specifically, an iterative procedure is used to derive $\beta_p(r)$ and $\alpha_p(r)$ vertical
profiles. A first-guess uniform LR profile of 38 sr (similar to the value used in the NASA-CALIPSO inversion at 1064 nm
for clean/polluted continental aerosol; Omar et al., 2009) is used in the first iteration step, and this  first guess is then updated
at each iteration using the $\beta_p$ and $\alpha_p=\alpha_p(\beta_p)$ profiles (i.e., the  modelled functional relationship) until the $\beta_p$ profile stabilises
(i.e., the difference of the vertically integrated aerosol backscatter, IAB, in two successive iterations is < 0.0025 m-1 sr-1).
A way to check the performance of the ALC-based optical properties retrievals is to compare the relevant tropospheric
integral of the aerosol extinction to the columnar AOD measured by an independent, co-located photometer. Within
Alicenet, these comparisons were performed over both short- and long- term datasets thanks to some co-located AERONET
(https://aeronet.gsfc.nasa.gov/, last access: 7 March 2024) or SKYNET (https://www.skynet-isdc.org/, last access: 7 March
2024) sun-photometers within the network. Specific examples are shown in Figs. 6 and 7, respectively.





Figure 6a shows the aerosol extinction profiles derived from the Rome-Tor Vergata ALC during the EMERGE-EU field
campaign in July 2017 (Andrés Hernandez et al., 2022), while in Fig. 6b the corresponding ALC-derived AOD is compared
to the same quantity from a co-located AERONET sun photometer in Rome-Tor Vergata.

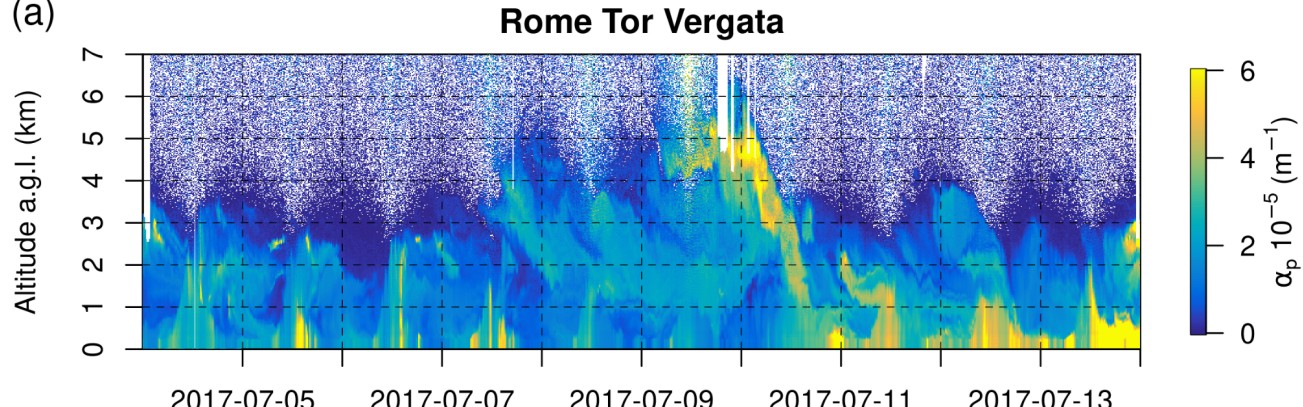

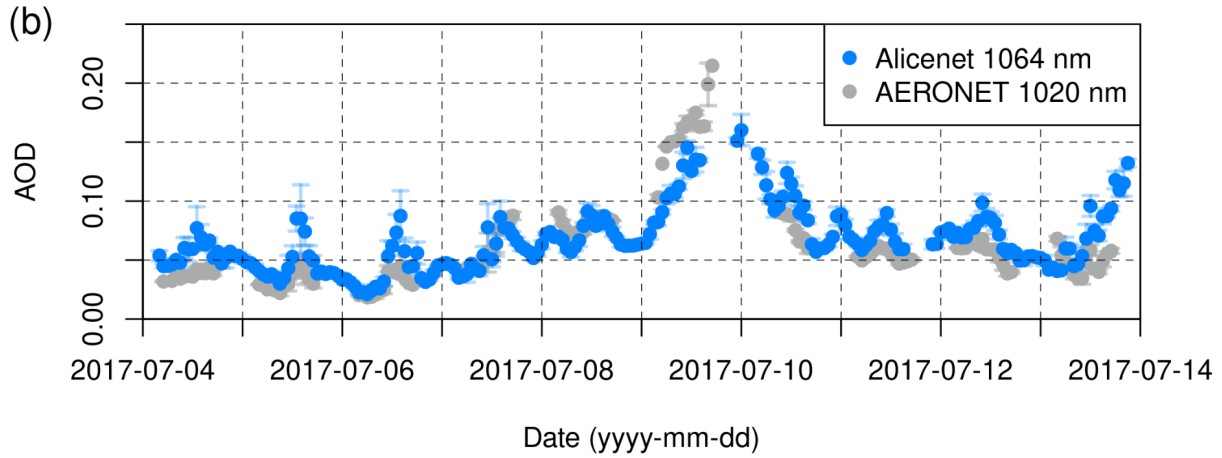

**Figure 6:** (a) Aerosol extinction profiles in Rome-Tor Vergata retrieved by the ALC Alicenet inversion during the EMERGE campaign in
July 2017, and (b) comparison between the relevant Alicenet- AOD and the co-located AERONET one (L2 data). Both Alicenet and
AERONET AODs are hourly averaged (error bars are the AOD standard deviations within this averaging interval).

The time series of the two independent datasets (both averaged at an hour resolution) agree within the expected uncertainties
(for AERONET, refer to Giles et al., 2019; for Alicenet, see Sect. 3.3.3). Some underestimation of the Alicenet retrieval is





found during the dates affected by transport of Saharan dust (e.g., 9 July 2017). This is expected because, as mentioned, the
functional relationship employed in the inversion was optimised for a continental-type aerosol and does not properly
describe the aerosol-to-extinction relation in desert-dust conditions (e.g., Barnaba and Gobbi, 2001). The extension of the
Alicenet retrieval approach to other aerosol types and relevant testing is planned for the future, also taking advantage of the
depolarisation measurements capabilities of the Vaisala CL61 operational within the network.
A more extended AOD comparison is shown in Figure 7, this exploiting  a 7-year long (2016-2022) and multi-site (Aosta,
Roma, Messina) ALC and sun-photometers co-located dataset. In particular, AOD data were averaged at 15-min resolution
and matched in time (i.e., time differences < 5 min). In Aosta, the SKYNET AOD was derived taking into account the
temperature correction of the POM-02 photometer as described in Uchiyama et al. (2018). Figure 7 shows that the Alicenet
retrieval is able to quantify the actual aerosol load in a variety of conditions. SomeALC overestimations are mainly due to
instrumental noise in the upper troposphere, while underestimations are mainly related to the presence of non continental
aerosol types. This is better illustrated in the Supplement S3 (Fig. S3), where the same results are shown highlighting data
associated with Angstrom Exponents < 0.5 (as derived from the co-located photometers), associated to coarse-mode
dominated aerosol types, as desert dust or sea-salt particles.

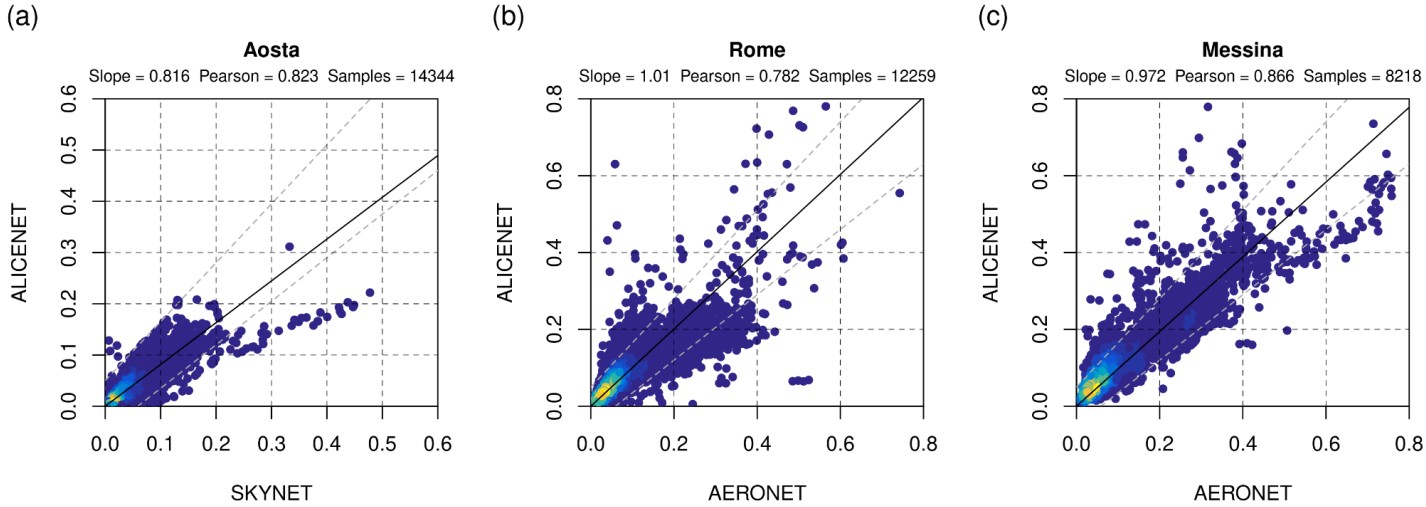

**Figure 7:** Comparison between the AOD derived by Alicenet (at 1064 nm) and AERONET/SKYNET sun photometers (at 1020 nm) in (a)
Aosta, (b) Rome Tor Vergata, and (c) Messina over the 2016-2022 dataset. Colours refer to the data density. Black line is the linear fit. Fit
slope and Pearson's correlation coefficients are reported in each panel together with the total number of data pairs (samples). Gray dashed
lines indicate a AOD-Alicenet deviation to the 1:1 line of  $\pm 0.01 \pm 0.15 \times AOD_{sunphotometer}$).




### 3.3.2 Aerosol physical properties

An approach similar to the one used in the retrieval of the optical properties (Sec 3.3.1) is also used to derive aerosol
physical properties. In fact, in the previous work by Dionisi et al. (2018) functional relationships linking the aerosol
backscatter to the particle surface area and volume ($S_p = S_p(\beta_p)$, $V_p = V_p(\beta_p)$) for a continental-type aerosol are also provided.
Aerosol mass concentrations ($M_p$) can then be derived from the estimated aerosol volume as $M_p = \rho_p\, V_p$, using an a-priori
aerosol density $\rho_p$ value. To evaluate the performances of the aerosol physical properties retrievals, the Alicenet estimates of
aerosol mass have been compared to in situ data. It is worth highlighting that remote sensing aerosol retrievals provide
aerosol properties in 'unperturbed' atmospheric conditions, i.e., including hygroscopic effects. Conversely, most in-situ
instrumentation (as those operating in AQMN in compliance to the EU AQ Directive) generally operates after drying the
aerosol samples. Therefore a RH 'adjustment' of the ALC-based aerosol properties is necessary when the two are to be
compared (e.g. Barnaba et al., 2010). For PM estimates, taking the in-situ (dry) measurement as reference, the ALC-derived
'wet' mass ($M_p$) can be corrected to 'dry' aerosol mass ($M_p^{dry}$) taking aerosol hygroscopicity into account.
In Alicenet, the dry aerosol mass is estimated following Adam et al. (2012):

$$M_p^{dry} = \frac{M_p^{dry}}{1 + 1/\rho_d\left(GF^3 - 1\right)} \qquad (4)$$

where

$$GF = \left(1 - \frac{RH}{100}\right)^{-\gamma} \qquad (5)$$

is the hygroscopic growth factor. The dry aerosol density $\rho_d$ and the $\gamma$ exponent depend on the aerosol particles under
investigation.
Here, both a short- (Fig. 8) and long- (Fig. 9) term comparison of the $M_p$ retrieved by Alicenet with reference measurements
are reported. The ALC datasets are from Aosta, which is one of the most tricky environments within Alicenet to deal with
aerosol hygroscopic effects.
In the first case, $M_p$ values at 3500 m a.s.l. (i.e., well above the region of incomplete ALC overlap ) extracted from the ALC
aerosol profiles are compared with in-situ aerosol concentrations measured by OPC at the same altitude ('Testa Grigia'
observatory, Plateau Rosa, 35 km-East of Aosta, see Supplement 4, Fig. S4) in June 2022. This period was selected because
during summer aerosols (mostly secondary hygroscopic particles) from the Po Basin are regularly transported to the western
Alps in thick layers, reaching altitudes > 4 km a.g.l. (Diémoz et al., 2019 a,b) thus likely impacting the two sites in a similar
way. In fact, June 2022 registered both long-range transport of desert dust and medium-range transport of Po Valley
pollution to Plateau Rosa. For the comparison, the ALC-based mean aerosol volume in the vertical range 3500 ± 200 m was



considered. A constant aerosol density of 1.5 g cm$^{-3}$ was used to derive $M_p$ values from both OPC- and ALC-based volume
concentrations. The ALC hygroscopic correction was only applied when the co-located PLC registered a linear volume
depolarisation ratio $\delta_v < 15\%$ within this altitude range, thus avoiding the irregular and poorly hygroscopic dust particles. For
these particles, the $\gamma$ exponent in Eq. 5 was setted to 0.2, as derived by D'Angelo et al. (2016). The modelled RH profiles
from MERIDA (Bonanno et al., 2019) were used in Eq 4.
Figure 8 shows the two mass estimates to exhibit a similar time evolution, with good agreement both in low aerosol
conditions (e.g. 6-15 June 2022), and during transport events increasing the local aerosol load, such as desert-dust intrusions
(e.g., 3-5, 18-22, and 27-28 June 2022) or Po Valley pollution advections (e.g., 13-14, and 25-26 June 2022). This result is
very promising, considering the horizontal distance between the ALC-probed column and the Plateau Rosa station (> 30 km,
Figure S4.2), and that the in-situ OPC measurements may also be influenced by local surface dynamics and emissions.

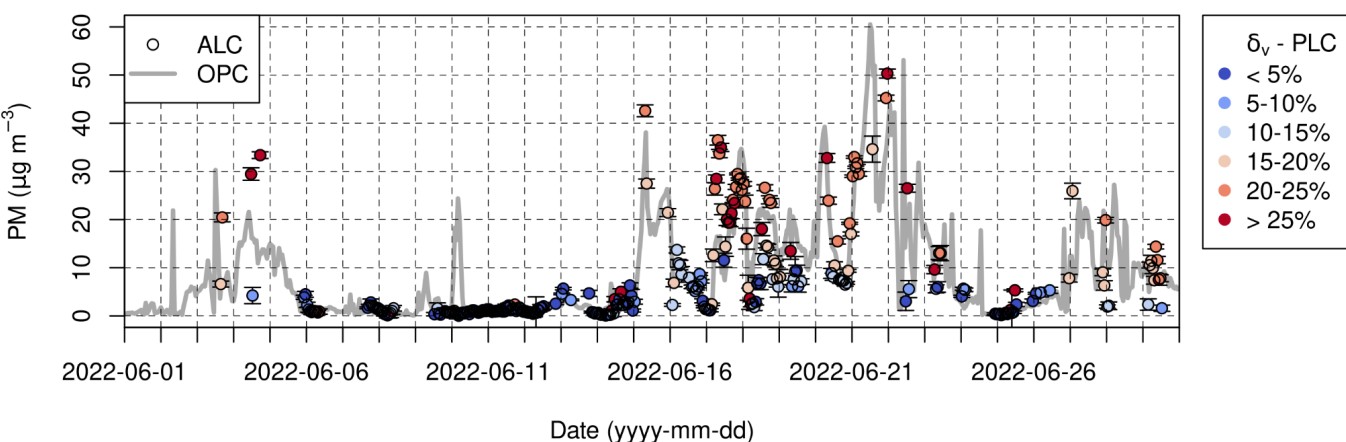

**Figure 8:** Aerosol mass concentrations derived from the Aosta ALC signals in the vertical layer 3500 ± 200 m a.s.l. (bullets, colour code
relates to depolarisation values from the co-located PLC), compared with the in-situ PM10 concentrations derived from OPC
measurements (grey line) at the mountain observatory 'Testa Grigia' ( Plateau Rosa, 3500 m a.s.l.) in June 2022 (OPC data courtesy of
Stefania Gilardoni, CNR-ISP, see also Figure S4 in the Supplement for details on site locations).

In Fig. 9, a 1-year (2021) comparison was performed using the Alicenet aerosol mass product $M_p$ and the in situ, surface
$PM_{10}$ concentrations derived by OPC measurements in Aosta downtown (4 km from the Aosta-Saint Christophe ALC;
Diémoz et al., 2021). The aerosol density was set to 1.5 g cm$^{-3}$, and the ALC hygroscopic correction was applied using
surface-level RH measurements. Figure 9 shows the daily median values, and corresponding 25-75 percentiles, retrieved
from both datasets. As can be observed, the Alicenet retrieved $M_p$ is able to reproduce the variability of the in-situ measured



PM$_{10}$, with some underestimations in the winter months. These underestimations are mainly attributable to i) the shallow
(i.e., few tens of metres), frequent temperature inversions occurring during winter in the Alpine valleys and capping aerosols
in the lowermost levels (Giovannini et al., 2020), and ii) the higher wintertime local emissions in the urban site of Aosta
downtown with respect to the semi-rural site where the ALC is operating (Diémoz et al., 2019b).

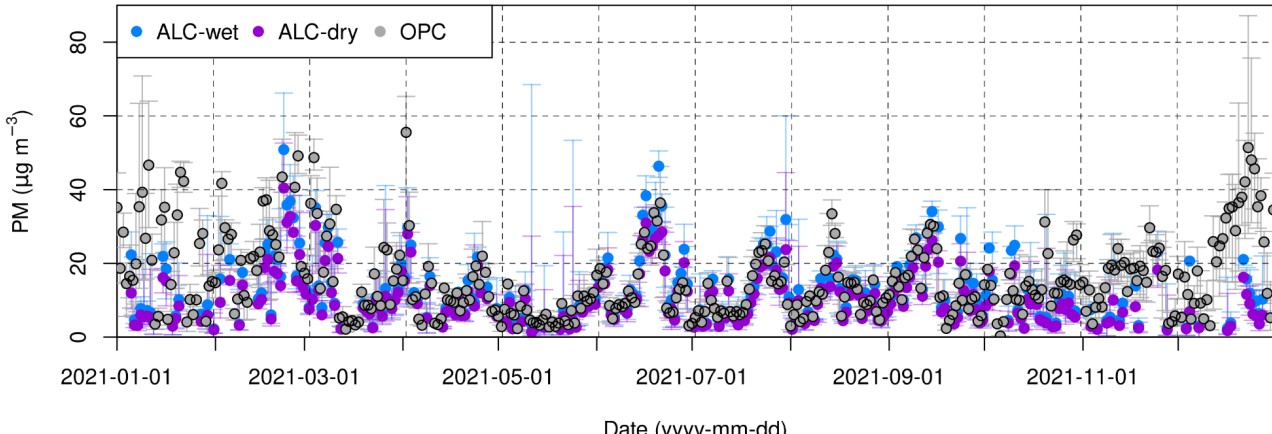

**Figure 9:** Daily median values (and relevant 25-75 percentiles, vertical bars) of a one-year (2021) dataset of aerosol mass concentrations
as derived by the Alicenet ALC inversion (first vertical level of the ALC signal) and by in situ OPC measurements in Aosta.

**3.3.3 Uncertainty of retrievals**
Despite the efforts described above to exploit the great potential of ALC to retrieve quantitative data for a series of aerosol-
related geophysical parameters, and the good performances of the retrievals shown above, due to several factors, associated
uncertainties to the Alicenet-derived parameters keeps of the order of 30-40%. In fact, the main sources of uncertainty of the
retrieved aerosol properties are associated with: 1) the instrumental noise of the signal, 2) the overlap correction applied to
the signal, 3) the variability of the instrument calibration coefficient, and 4) the accuracy of the functional relationships in
terms of both internal uncertainty and their representativeness of the actual aerosol type sounded. The first factor depends on
the instrument status and impacts the retrieval mainly in the middle-upper troposphere. The second factor, impacting in
particular the lowermost levels, depends on the accuracy of the overlap correction, i.e., on the statistical and physical
representativeness of the ensemble of overlap functions from which the overlap model is derived (Sect. 3.1.2). The third
factor, derived as outlined in Sect. 3.2, directly impacts the accuracy of $\beta_{att}$. For example, it is found by propagation that





changes of 30% in the instrument calibration coefficient (which are quite usual in some Alicenet and E-PROFILE stations)
may result in errors in $\beta_{att}$ up to 20%. The fourth factor, impacting the accuracy of $\alpha_p$, $V_p$, and, to a lesser extent, $\beta_p$, strongly
depends on the actual aerosol conditions: the functional relationships used can give a good estimate of the aerosol properties
in presence of continental aerosols, while in presence of non-continental particles they are less accurate (a relative error of
30-40% was derived by Dionisi et al., 2018). As mentioned, extension of the Alicenet approach to include other aerosol
types is foreseen for the next future, particularly exploiting the PLC depolarisation profiles for aerosol-typing, and thus
driving selection of aerosol-type specific functional relationships (e.g. Gobbi et al., 2002).
Concerning the retrieval of aerosol mass concentrations, the assumed particle densities are a major source of uncertainty, and
the accuracy of the retrieval depends on the possibility to better constrain  the aerosol density profiles, e.g., through ancillary
data, including depolarisation information.
Overall, the above factors result in time and range-dependent uncertainties of the ALC-based aerosol optical and physical
properties. Their amplitude and vertical variability depend on the instrument characteristics (e.g., amplitude and shape of the
signal noise), and on the actual aerosol composition and stratification. The  expected uncertainty with an optimal SNR up to
at least 7 km a.g.l., an overlap error < 10% in the lowermost levels, and in presence of continental aerosol types is of 30-40%
for the AOD, reaching 50% for aerosol mass.
**3.4 Automatic identification of aerosol layers**
In addition to the quantitative retrievals of specific aerosol properties, ALC measurements can also provide a simplified yet
useful information on a key aspect of atmospheric studies, i.e., vertical stratifications, from the lowermost levels to the mid-
upper troposphere, using aerosols as tracers.
Due to the influence of the Earth surface, in the lower troposphere the atmospheric stratification generally features a
temperature inversion, capping locally emitted aerosols, trace gases and moisture. During daytime, the heating of the Earth
surface triggers the development of turbulent fluxes, mixing the above quantities within the so-called Mixed Boundary Layer
(MBL) and causing its growth by entrainment of upper air up to the capping inversion (Stull, 1988). In the mid-upper
troposphere, atmospheric circulations drive large scale aerosol transport processes. The aerosol loaded air masses are
generally advected above the MBL, but may descend and be entrained within it. It is worth mentioning that the aerosol
stratification depends not only on local (mainly vertical) and synoptic (mainly horizontal) scale dynamics, but also on
(horizontal and vertical) mesoscale circulations, in particular over complex terrain (Serafin et al., 2018; Collaud Coen et al.,
2018). Moreover, stagnation, secondary aerosol formation, hygroscopic effects, and photochemical reactions can
significantly modulate aerosol concentrations within specific atmospheric layers (Collaud Coen et al., 2011; Curci et al.,
2015; Sandrini et al., 2015) making this scheme more complex to be interpreted. However, three main aerosol layer types
can be generally identified:



a) a layer characterised by the continuous (in space and time) presence of aerosols (Continuous Aerosol Layer, CAL), as a results of a combination of factors, among which surface aerosol emissions, capping inversion effects, convection processes, secondary aerosol formation;

b) within the CAL, a layer where aerosols are mixed by surface-driven turbulent fluxes (Mixed Aerosol Layer, MAL), whose pronounced diurnal variability is mainly determined by local thermal forcings and, in certain sites, by mesoscale circulations transporting or removing particles in the lower troposphere;

c) aerosol layers detached from the surface (Elevated Aerosol Layers, EALs), above the MAL and either within or above the CAL, generally resulting from medium-to-long range advections, or from the development of the local residual layer during nighttime.

A novel, stand-alone procedure (ALADIN: Aerosol LAyer DetectIoN) was developed within Alicenet to extract and make this (CAL, MAL, and EAL) layering information usable within the network. This procedure is described hereafter, with examples of the products over the short and long term. It can be applied on both ALC and PLC.

A variety of methodologies have been developed to derive quantitative information on the aerosol layering from ALCs (e.g., Haeffelin et al., 2012). These are based, for example, on the Continuous Wavelet Transform (CWT) analysis (Morille et al., 2009; Caicedo et al., 2020) or the detection of anomalies with respect to background aerosol conditions (Adam et al., 2020). Methods based on the variance and gradient analysis of ALC signals (e.g., Angelini et al., 2009; Poltera et al., 2017; Kotthaus et al., 2020) or image processing techniques (e.g., Vivone et al., 2021) have also been applied for the detection of the MBL height. It is worth mentioning that the MBL and MAL definitions should not be considered synonyms, since the first is based on thermodynamic quantities, while the second uses aerosols as tracers.

Within the Alicenet ALADIN procedure, the CAL height is derived from cloud-screened, denoised, and calibrated ALC profiles averaged at 30 min resolution. It is simply defined as the altitude of the layer in which $\beta_{att} > \beta_{mol}$ for at least 98% of its extension.

The MAL is identified through a technique disentangling regions where aerosols are mixed by (vertical) turbulent fluxes from the ones where they are transported by (mainly horizontal) large-to-medium scale circulations. More specifically, the Dynamic Time Warping algorithm (DTW, Giorgino et al., 2009) is applied to a sequence of denoised $\beta_{att}$ profiles at 1-min resolution. In brief, this algorithm computes the local stretch or compression to be applied to each profile in order to optimally map the preceding into the following. Its output field ($w_{DTW}$) can be interpreted as the local vertical displacement of the aerosol-loaded air parcels, while the region near the surface where $w_{DTW}$ rapidly changes in sign and magnitude as the region where the mixing is acting. In order to select this region, the standard deviation of $w_{DTW}$ ($w_{SD}$) is calculated over 30-min intervals (as generally done in Eddy-Covariance analysis; Aubinet et al., 1999) and the local minima along the $w_{SD}$ profiles are identified. The MAL height is then defined as the height of the first $w_{SD}$ local minima verifying specific criteria (details are provided in the Supplement, Sect. S5). It is derived only when ALC profiles below 500 m a.g.l. are cloud-free.

The identification of the EALs is based on the CWT analysis followed by an iterative technique. More specifically, the identification of the layer 'centre' takes advantage of the CWT algorithm (MassSpecWavelet R package) developed by Du et



al. (2006), which is applied to cloud-screened and denoised $\beta_{att}$ profiles at 30-min resolution. This algorithm exploits both the
CWT and $\beta_{att}$ coefficients to identify peaks attributable to aerosol layers and discriminates these from noise. Once an aerosol
layer 'centre' is identified, its top and bottom boundaries are determined with an iterative procedure. More specifically, a
grid of 'potential' bottom and top boundaries is constructed, and for each boundaries combination an ad hoc metric is
calculated (it takes into account the deviation of the $\beta_{att}$ profile with respect to a reference profile ($\beta_r$), the integrated $\beta_{att}$
coefficients, and the CWT coefficients). The EAL is then identified as the window maximising this metric (details, including
quality controls, are provided in the Supplement, Sect. S5). In this approach, the choice of the reference profile $\beta_r$ to be used
depends on the intended application: molecular attenuated backscatter profiles are used to detect aerosol layers with respect
to a clean atmosphere, while 'climatological' profiles (e.g. site- and monthly-dependent median $\beta_{att}$ profiles derived from
Alicenet multi-annual datasets) are used to identify aerosol layers representing anomalies with respect to typical aerosol
conditions. Mean properties of the EAL are also derived, and namely, the mean AOD, estimated using the aerosol extinction
profiles from CHM15k systems, or the mean fraction of irregular (mainly dust) particles, estimated from the linear volume
depolarisation ratio ($\delta$) values of CL61 systems as follows (Tesche et al., 2009):
$$F_d = \frac{\overline{(\delta - \delta_{nd})(1 + \delta_d)}}{(\delta_d - \delta_{nd})(1 + \delta)} \qquad\qquad (6)$$

where $\delta_d$ and $\delta_{nd}$ are the typical dust and non-dust volume depolarisation values (assumed as 0.25 and 0.05, respectively), and
the overbar denotes the median within the bottom and top boundaries of the layer.
Figure 10 shows an example of the 'layering mask' derived from the overall ALADIN procedure, this referring to the same
CHM15k data (Rome-Tor Vergata site) shown in Fig. 6. The mask discriminates between the vertical ranges characterised
by the presence of aerosol layers (CAL, MAL, EALs), aerosol-free (i.e., molecular, MOL), and cloud-screened (CLOUD)
regions. In this case, the EALs are identified using as reference profile ($\beta_r$) the median summertime $\beta_{att}$ profile derived from
the multi-annual (2016-2022) dataset of the instrument. In the episode reported in Fig. 10, the elevated layers above 3 km
a.g.l. are mostly due to Saharan dust advections, while the ones below 2-3 km a.g.l. to the development of the residual layer
during the night (e.g., 5-6 July 2017), or the presence of fire plumes travelling in the lower troposphere (e.g., 11 July 2017).
This further discrimination was made possible through the analysis of the depolarisation profiles of the prototype-CHM15k
PLC in downtown Rome (not shown), and ancillary information (models, satellite). The inclusion of the CL61 depolarisation
information directly within the layering procedure as outlined above represents a first step to automate the layer typing
capacity within the network (e.g., Nicolae et al., 2018; Córdoba-Jabonero et al., 2018).



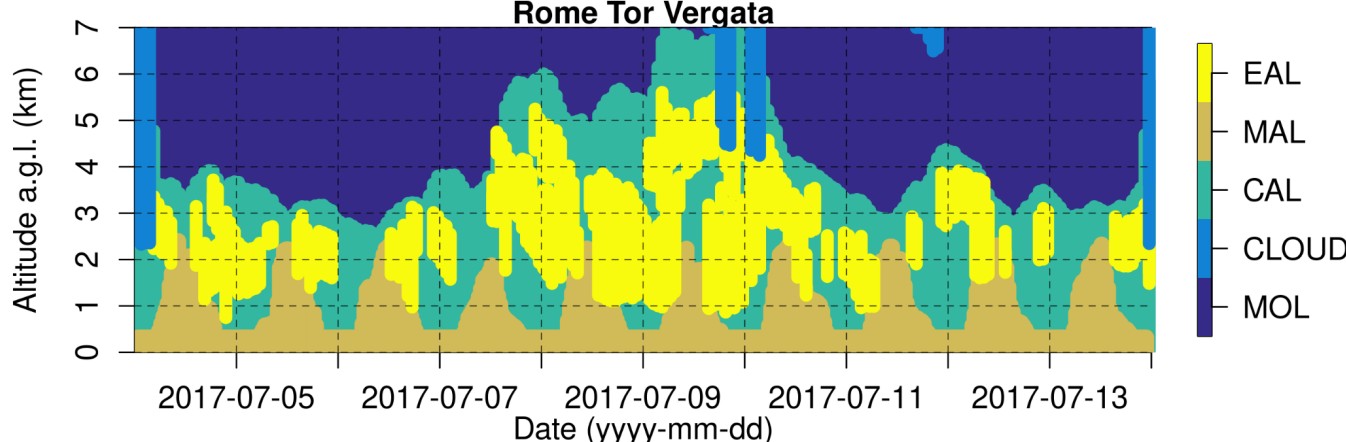

**Figure 10:** Atmospheric layering mask derived from the Alicenet ALADIN processing on the CHM15k operating in Rome - Tor Vergata
in the same period presented in Fig. 6. The mask discriminates the following layers: the continuous aerosol layer (CAL), the mixing
aerosol layer (MAL) and elevated aerosol layers (EALs), as well as aerosol-free, molecules-only regions (MOL) and clouds-affected
(CLOUD) vertical ranges.

An example of the results derived from the application of the ALADIN procedure on a multiannual (2016-2022) ALC
dataset is given in Fig. 11. This shows the monthly-resolved statistics (median and 25-75 percentiles) of the CAL (cyan) and
MAL (green) daily cycles derived from the CHM15k dataset in Rome Tor-Vergata. The CAL parameter clearly shows an
yearly cycle (minimum in winter and maximum in summer) while its daily cycle is slightly different in winter (thicker
during the night, likely due to enhanced hygroscopic growth of particles) with respect to summer (thicker in the mid part of
the day, likely driven by enhanced convection). As expected, all over the year, the MAL shows a marked daily cycle, with
maximum heights in summer (about 2 km thick in July-August) doubling those in winter (about 1 km in December-January).
For direct comparison to the ALC-derived, aerosol-traced MAL, similar monthly- and daily- resolved statistics of turbulent
kinetic energy (TKE) data were also derived, using a co-located ultrasonic anemometer (magenta lines). Note that,
differently from the other Figures, in which time was always reported as UTC, in this case Central European Time (CET) is
used to better highlight the diurnal (a.m./p.m.) variability of the quantities addressed. As can be observed, the timing of the
moment the MAL starts increasing during the morning matches the TKE one, although the former grows faster, anticipating
its daily maximum with respect to the TKE one. This temporal shift is kept during the afternoon, especially in summer. This
is likely due to the fact that during the morning the ALC signal increases due to an increase of local aerosol emissions, and
not just due to turbulent energy-driven convection. In the opposite way, the typical land-sea breeze developing in Rome in



the early afternoon (Di Bernardino et al., 2022) acts in removing aerosol particles, thus leading to a decrease of the ALC-based MAL estimation, while enhancing the mechanical component of the TKE, which in fact starts decreasing later in the day.



**Figure 11:** Monthly and daily resolved statistics of the MAL and CAL heights (median, and 25-75 percentiles as dashed areas) derived from the CHM15k in Rome Tor Vergata (2016-2022 dataset), with the relevant TKE statistics derived from a co-located ultrasonic anemometer (violet line and dashed area).

A more detailed long-term and Alicenet multi-site analysis of aerosol properties and stratifications is in progress (Bellini et al., 2024a).

**4. Potential from 4D near-real time aerosol monitoring**

The continuous monitoring capability of the Alicenet network has been already exploited in past events to characterise specific aerosol transport features and/or to quantify the impact of aerosol dynamics on local aerosol concentrations, mostly in synergy with other tools and measuring techniques as in-situ aerosol observations, ground-based passive remote sensors, satellites or models (Gobbi et al., 2019; Diémoz et al., 2019a,b; Rizza et al., 2022; Tositti et al., 2022, Andres Hernandez et al., 2022). This section describes, through some recently recorded showcases, the potential of the near real time Alicenet monitoring at the national scale, that could also represent a  useful tool for nowcasting/warnings/alerts in case of noteworthy events.

**4.1 Po Valley local dust front (14 April 2020)**





In a previous study (Diemoz et al., 2019a,b), the operational use of Alicenet provided observation-based evidence of the
export of pollutants from the Northern Italy Po Valley to surrounding areas. The phenomenon, previously observed by lidar
profiling performed at the EC-JRC in Ispra (about 60 km northwest of Milan, Barnaba et al., 2010), was further analysed
and quantified thanks to the Alicenet combination of sites (within and at the border of the Po Valley), demonstrating that
such advections markedly affect PM-related AQ even in the 'pristine' mountain environments mainly transporting
hygroscopic particles of secondary origin. However, transport of particles of primary origin (particularly from soil-related
sources) across the Po Valley has been also observed, particularly during dry periods. Figure 12 shows an example of such
events (14 April 2020), largely impacting regional AQ and visibility.

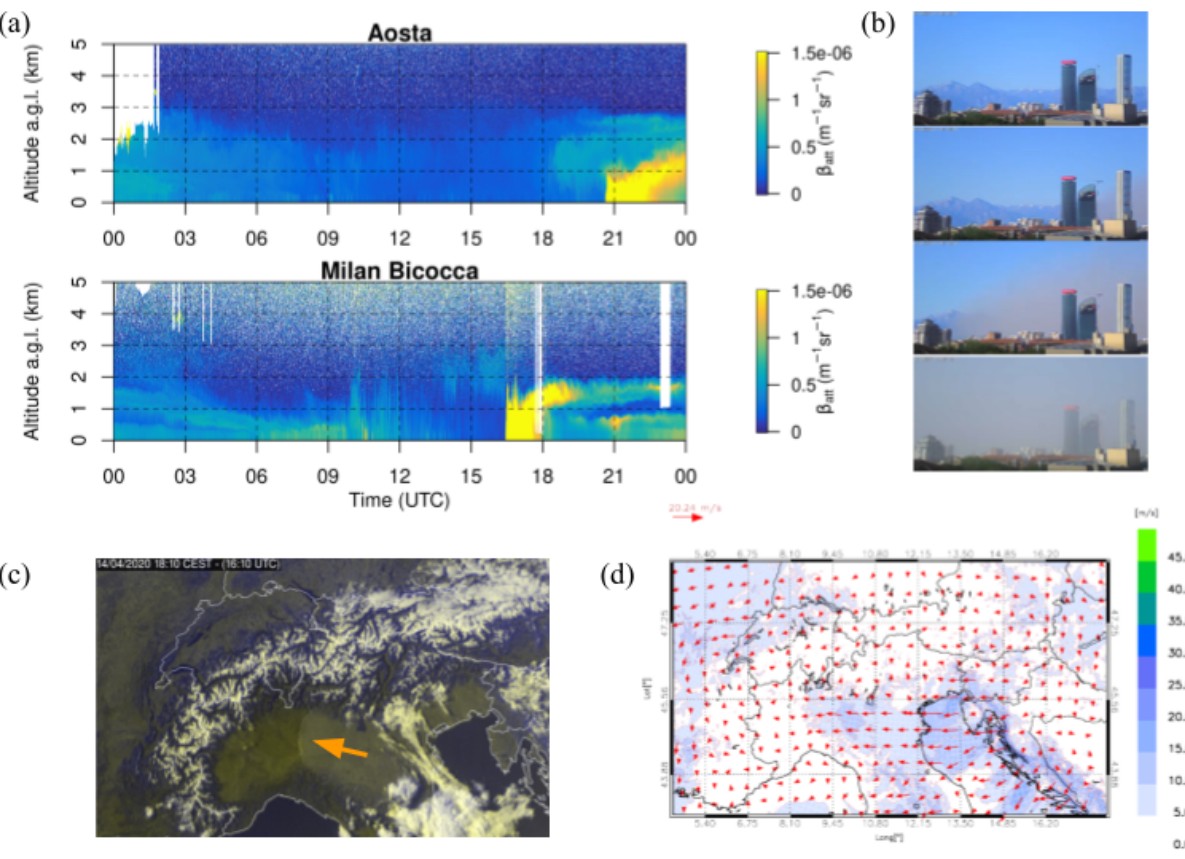

**Figure 12:** (a) Total attenuated backscatter profiles at Aosta and Milan-.Bicocca sites on 14/04/2020; (b) central Milan webcam (Source:
Arzaga meteorological observatory, https://www.osservatorioarzaga.it/) showing the rapid decrease of visibility on 14/04/2020 (from top
to bottom: 16:08, 16:15, 16:20, 16:25 UTC), (c) Po Valley satellite true colour image (14/04/2020 18:10 UTC; Credits: EUMETSAT) with
indication of the regional dust front (orange arrow), and (d) 10 m wind speed and direction simulated by WRF over North Italy



(14/04/2020 17:00 UTC, data courtesy of Stefano Federico CNR-ISAC) illustrating the extension of the gust and wind fronts. The arrival
of the dust front in Milan at 16:20 UTC and in Aosta at 20:40 UTC is clearly visible from ALC profiles.

In fact, anomalous dry conditions affected Europe in April 2020. ALC profiles in Aosta and Milan-Bicocca (Fig. 12a) clearly
capture the timing of the plume's arrival in Milan (as also seen from central Milan webcams, Fig. 12b) and show the vertical
extent of the particle-rich layer associated with the episode. The plume continued to travel westward and was detected by the
ALC in Aosta 4 hours later, indicating a wind speed > 12 m/s. As revealed by both satellite measurements (Fig. 12c) and
model simulations (Fig. 12d), this episode was due to an extended (about 100 km) gust front originating from the cold and
intense Bora winds from East, resuspending and transporting soil-originated particles from the cultivated fields across the
whole Po Valley.
**4.2 Advection of Saharan dust and Canadian fire plumes over Italy (19-28 June 2023)**
The Mediterranean area is frequently affected by the transport of desert dust from North Africa and the Middle East (e.g.,
Barnaba and Gobbi, 2004; Querol et al., 2009; Basart et al., 2012a; Greilinger et al., 2019; Gama et al., 2020). In Italy, these
events are estimated to reach the ground on 10% (Northern regions) to over 30% (Southern regions) of the days in a year,
and to impact on surface daily-mean PM10 concentrations with 10-15 µg/m$^3$ (Barnaba et al, 2022). The transport of fire
plumes from global-to-medium distances is also an important contributor to surface PM concentrations. Concerning the
global scale transport towards Europe, a significative contribution is given by forest fires regularly developing during boreal
summers in Canada (e.g., Ceamanos et al., 2023; Shang et al., 2024), although a major contribution from agricultural fires in
Europe has also been detected over the continent, particularly in spring and summer (Barnaba et al., 2011). Last summer
(2023) was particularly impacted by multiple episodes of severe wildfires in central Canada. Almost 480 megatonnes of
carbon were emitted, resulting in a major impact on AQ across Canada and the Northern US. The plumes have also been
observed to be regularly transported towards Europe (https://atmosphere.copernicus.eu/copernicus-canada-produced-23-
global-wildfire-carbon-emissions-2023, last access: 6-3-2024). Figure 13 shows a composite of measurements collected at
multiple Alicenet sites across the country during a 10-days period (19-28 June 2023) affected by both desert-dust (time-
range orange boxes) and forest-fire plumes (time-range magenta boxes). More specifically, this period was characterised by
the intrusion of Saharan dust to Southern to Northern Italy (19-24 June 2023), followed by the transport of Canadian fire
plumes over Central and Northern Italy (27-28 June 2023). The ALC profiles ($\beta_{att}$ and $\delta_v$) at the 7 selected Alicenet sites
allow to follow the spatio-temporal evolution of the different aerosol layers and identify the relevant aerosol type. The
Saharan dust layers were firstly observed over South-West Italy (Capo Granitola, June 19 in the morning), then moving
westward to Messina and Catania Nicolosi (June 19, afternoon), and northward to Turin, Aosta, Milano, Mt. Cimone, where
the dust plume is detected in the evening. All over Italy, the dust plume affects atmospheric layers up to 7 km altitude,
reaching down to the surface on June 20. In fact, the PLC systems clearly indicate the presence of irregularly-shaped mineral





particles (depolarisation values > 30%), these mixing with local (mainly spherical) particles ($\delta_v$ ~10-20%) when reaching the
lowermost levels.

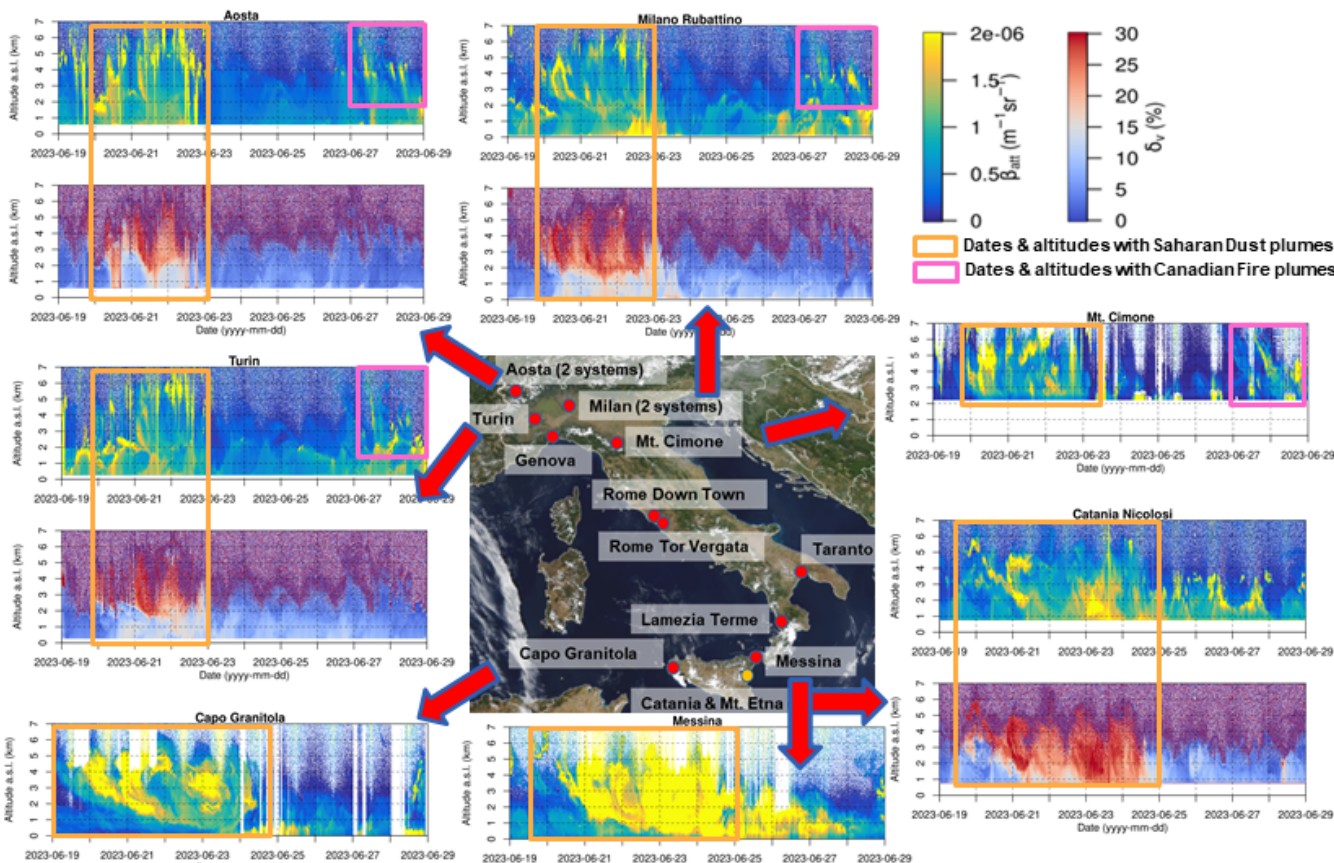

**Figure 13:** Vertical profiles of total attenuated backscatter, $\beta_{att}$ (ALCs & PLCs) and volume depolarisation, $\delta_v$ (PLCs) for selected, North-
to-South Alicenet sites in the period 19-28 June 2023 affected by Saharan dust and Canadian fire plumes (orange and magenta boxes,
respectively). Central map: satellite true colour image (credits: EUMETSAT).

The Canadian fire plumes were firstly observed by Alicenet systems operating in Northern Italy on 27 June 2023, impacting
atmospheric levels in the range 2-7 km a.s.l. The plumes were firstly detected in Aosta, and then travelled through the whole
Po Valley, being clearly observed in Mt. Cimone. Being mainly composed of processed particles, these long-range
transported fire plumes do not show increases of depolarisation values, and appear as thinner aerosol layers with respect to
the ones typically associated with dust layers. These vertically resolved measurements well complement the information that
can be gathered from satellites as shown by this comparison of Alicenet data and MSG and Metop retrievals, limited to the
dust event (e.g., https://vuser.eumetsat.int/resources/case-studies/dust-transport-from-the-sahara-to-the-mediterranean, last
access: 6-3-2024). At the same time, this information also provides an observational verification of the picture that can be



obtained by modelling tools. In this respect, Fig. 14 shows the Ensemble CAMS EU forecast maps for two dates within the
temporal window addressed: 22/06/2023 (dust intrusion, left panels) and 27/06/2023 (Canadian fires, bottom panels), at two
altitude levels (100 and 3000 m a.g.l., top and bottom panels respectively). The horizontal evolution of the aerosol
advections qualitatively agrees with the information detected by Alicenet. It is more difficult to correctly model the aerosol
vertical distribution, due to both their coarse vertical resolution and simplified parameterizations of the aerosol-related
atmospheric processes (e.g., Koffi et al., 2016). Hence, remote sensing observations represent an added value, in particular
for AQ monitoring, needing an accurate description of the timing and load of the aerosol injections in the lowermost
atmospheric levels, and model validation exercises.

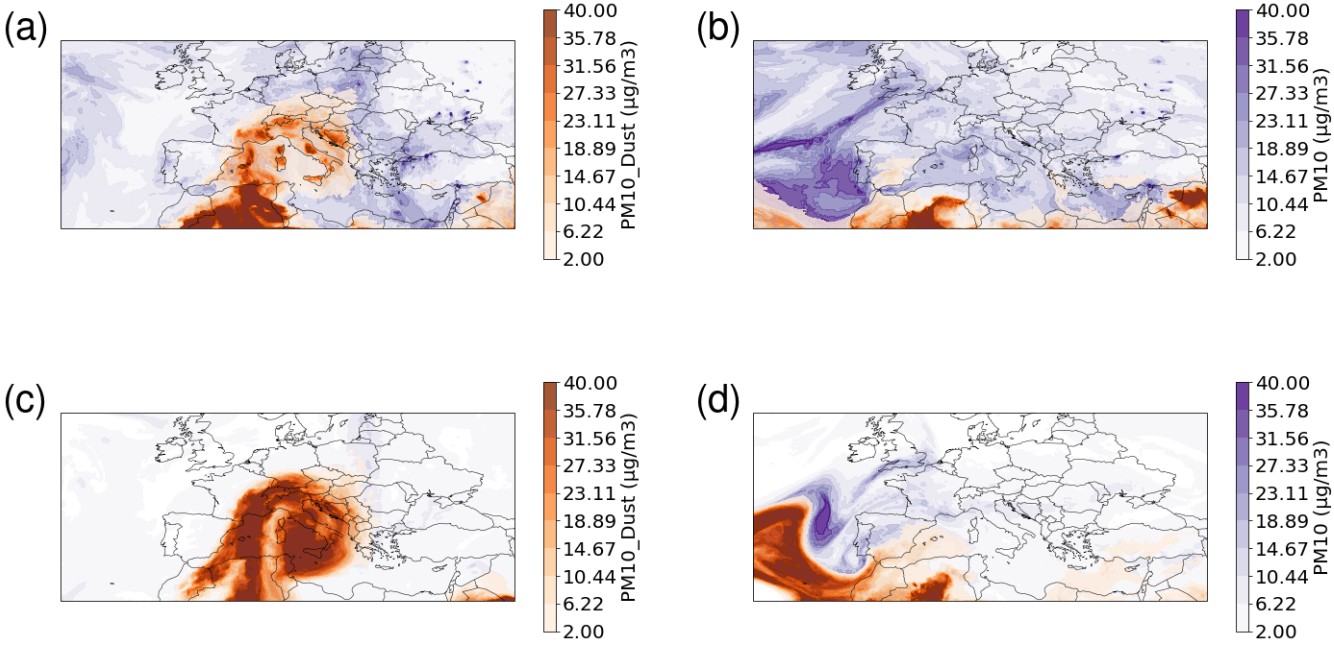

**Figure 14:** CAMS EU forecast of the total PM10 and PM10-dust component concentrations during the desert dust (22/06/2023 00:00 UTC
- left panels) and the Canadian fires (27/06/2023 21:00 UTC - right panels) events of Figure 13,  top (bottom) panels referring to  100 m
(3000 m) altitude.

**4.3 Aerosol particles from the Mt. Etna eruption (13-14 August 2023)**



A recent showcase from the Etna volcano eruption is reported here to highlight the important information that ALC/PLC
observations can provide in volcanic areas to complement in situ, satellite-based and modelling data (e.g., Corradini et al.,
2018, Scollo et al., 2019). During the night between 13 and 14 August 2023, this Europe's most active volcano erupted, its
Southeast Crater emitting a volcanic cloud that the PLC in Catania Nicolosi detected to reach up to 5km at 21 UTC (Fig
15a). On August 13, at 20:41 UTC, a Volcano Observatory Notice for Aviation (VONA) was issued by INGV
(https://www.ct.ingv.it/Dati/informative/vona/VONA_Etna_202308132041Z_2023005708E01.pdf, last access: 06-03-2024)
with a 'red alert' for aviation. VONA are short, plain-English messages aimed at dispatchers, pilots, and air-traffic
controllers to inform them of volcanic unrest and eruptive activity that could produce ash-cloud hazards. In fact, flights
serving Catania were halted. The most intense phase of the eruption occurred between 01:40-02:30 UTC, when PLC
depolarisation values reached values > 40% indicating a predominance of  irregular ash particles. The ash plume was then
observed to rapidly reach down to the ground, while moving southward in the Mediterranean Sea (Fig 15b). In fact, less than
5 hours after the beginning of the eruption the plume was detectable east of Malta. In agreement with the ALC record, the
VONA issued by INGV at 05:54 UTC indicates that no ash plumes were produced and that the volcanic ash was confined in
the  summit  areas  of  the  volcano,  this  corresponding  to  an  orange  Aviation  colour  code
(https://www.ct.ingv.it/Dati/informative/vona/VONA_Etna_202308140554Z_2023005808F01.pdf, last access: 06-03-2024).





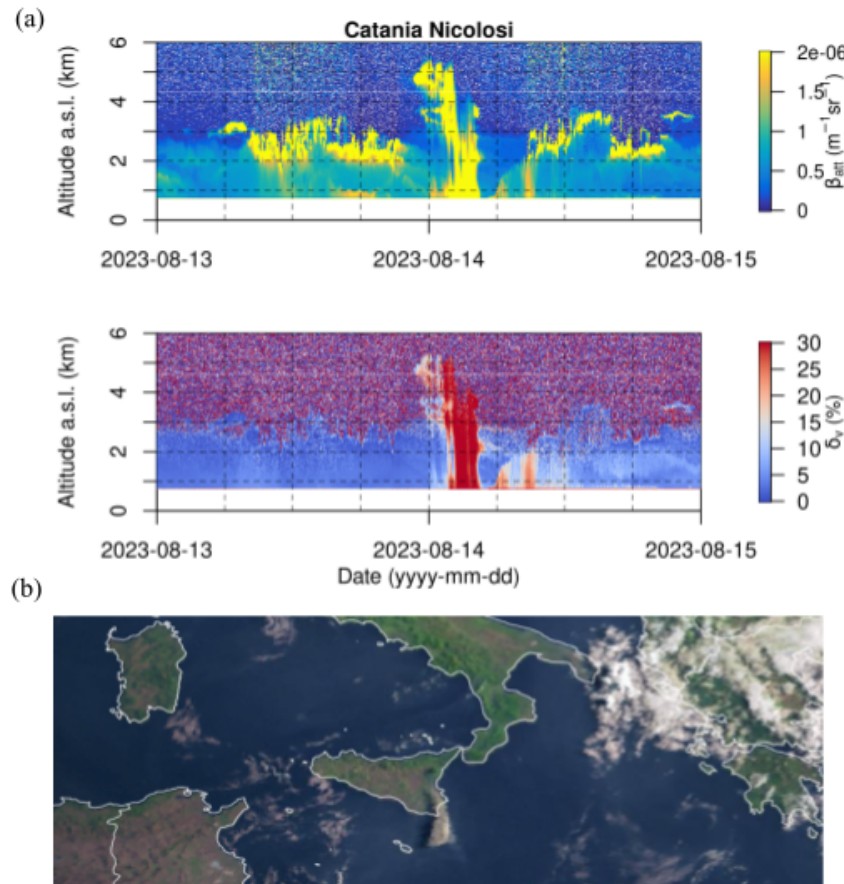

**F**igure 15: **(**a) Total attenuated backscatter, $\beta_{att}$, plus volume depolarisation, $\delta_v$, profiles observed at the Alicenet Catania Nicolosi site on 13-14/08/2023; (b) METEOSAT Natural Colour Enhanced RGB (SEVIRI) image referring to 14/08/2023, 05:15 UTC (Credits: EUMETSAT).

## 5 Summary and future perspectives

In this work we present Alicenet, the Italian network of automated lidar-ceilometers (ALCs) operating from North to South across the peninsula. It is a cooperative network set up by CNR-ISAC in 2015, with active contributions from several national and regional institutions (e.g. regional EPAs, Universities, Research Centres and private companies). Some Alicenet ALCs also contribute to the European network E-PROFILE managed by EUMETNET to fill an Italian observational gap at the EU level. In fact, in most Member States meteorological agencies are typically responsible for ALC monitoring, some of these running over 100 instruments (e.g. the German meteorological service, DWD,



https://www.dwd.de/EN/research/observing_atmosphere/composition_atmosphere/aerosol/cont_nav/aerosolprofiles.html,
last access 7-3-2024).
In recent years the Alicenet network grew up and now includes about 20 active systems (Table 1) sampling in very different
environments (urban, coastal, mountainous and volcanic areas), and thus providing information in a large spectrum of
atmospheric conditions and aerosol types. Alicenet promoted a standardisation of instruments and an homogeneous data
processing specifically developed within the network. It mainly runs single-channel ALCs (CHM15k systems by Lufft) but
is progressively introducing  polarisation-sensitive systems (PLCs) recently commercialised by Vaisala (CL61) to further
exploit the ability of these systems to discriminate among aerosol types. Alicenet also intends to bridge a gap at the national
level between the research-oriented and the operational use of active aerosol remote sensing devices in several sectors,
among which: a) air quality, b) radiative budget/solar energy, c) validation of models and satellite products, d) aviation
safety. In the first case, of particular interest are the abilities of the ALC/PLC-based Alicenet data to i) automatically identify
medium-to-long range aerosol advections and estimate the relevant contribution to surface PM10 concentrations, and ii)
provide continuous information on particulate matter layering, including the Mixing Aerosol Layer, i.e. on the atmospheric
volume in which locally emitted particles are diluted (e.g., Kotthaus et al., 2023). The effectiveness of using these ALC/PLC
abilities in support of standard air quality monitoring networks is being currently explored within the ongoing EU H2020
Project RI-URBANS, aimed at developing an air quality monitoring system that complements those currently available. In
this framework, tests of upscaling the Alicenet tools to other urban sites in the EU are in progress (Bellini et al., 2024b).
Concerning the other applications mentioned above, the continuous ALC-based information on the aerosol loads and vertical
distribution is useful to better estimate the relevant aerosol radiative effects (beneficial for example within an operational
short-term solar forecasting system based on a multisensor approach, e.g. Papachristopoulou et al., 2024), for validation
of/assimilation in models (e.g. Chan et al., 2018; Valmassoi et al., 2023), verification of satellite aerosol layering products
(e.g., Janicke et al., 2023) including those foreseen from the upcoming EarthCare ESA mission (e.g., van Zadelhoff et al.,
2023) or for the provision of near-real time alerts for aviation safety during specific extreme events such as desert dust
storms and volcanic eruption (e.g., Papagiannopoulos et al., 2020).
Since the beginning of the Alicenet activities, particular care has been devoted to data retrievals and exploitation, this also
taking advantage of technical/scientific exchanges within European initiatives, such as E-PROFILE and the EC Cost Actions
TOPROF (2013-2016) and PROBE (2019-2024). To this purpose, Alicenet developed a specific, centralised and automated
data processing chain and associated data quality assurance/data quality control (QA/QC) procedures which are presented in
detail in this work. Overall, the data processing includes signal correction and calibration procedures (Sects. 3.1, 3.2) and
retrieval algorithms for both quantitative aerosol properties (Sects. 3.3.1, 3.3.2) and aerosol layering (Sect. 3.4). The
processing output thus includes basic-to-advanced aerosol products (L1-L3 products), ranging from attenuated backscatter
and (if available) depolarisation profiles, to aerosol extinction and mass concentration profiles plus aerosol layering (e.g.,
Mixed Aerosol Layer and Elevated Aerosol Layers). Vertical profiles of L1 and/or L2 products are provided in near real time
on a dedicated website (https://www.alice-net.eu/, last access: 3 March 2024), while L3 products are routinely obtained



offline and are currently only available upon request. Examples of both product types are reported in Sect. 4 and Sect. 3,
respectively. For L3 products, this work also includes direct comparisons with relevant, independent data (in-situ or remote
sensing, depending on the variable addressed), this showing the Alicenet data processing to provide robust and quantitative
aerosol information, within the discussed limits of the data accuracy (Sect. 3.3.3). In fact, long-term comparisons of aerosol
mass retrievals with surface PM10 data show mean discrepancies of 35%, while AOD comparisons to thousands of relevant
data points from co-located sun photometers show correlation coefficients > 0.8 and fit slopes ranging between 0.8-1.0,
depending on the site location.
A more extended analysis of the L3 products multi-annual datasets at different Alicenet sites located across Italy, , will be
presented in a separate work (Bellini et al., 2024a), while an algorithm intercomparison exercise is currently in progress
within PROBE to evaluate differences in the outcomes produced by different national networks in the EU (namely: Alicenet
- Italy, MetOffice - UK, V-PROFILE - Norway, DWD - Germany).
Besides a geographical extension of the network (new stations are already planned to join in the next months), next steps
foreseen for Alicenet, are: a) a better characterisation of the instruments artefacts and calibration, b) the extension of the
Alicenet ALC retrieval methodology to different aerosol types, c) the development of a full retrieval for (CL61) PLCs,
further exploiting the depolarisation information to complement aerosol layering with aerosol typing. The CL61 operating at
a different wavelength with respect to CHM15k, this would require the evaluation of water vapour absorption corrections
(e.g., Wiegner and Gasteiger, 2015), and the definition of new, wavelength specific functional relationships (e.g. Dionisi et
al., 2018) to be used within the data inversion process. The feasibility of a routine dissemination of Alicenet L3 products via
the network website in addition to the near-real time L1 and L2 ones is currently under evaluation.
Overall, Alicenet represents a valuable resource to complement the aerosol observational capabilities in Italy with the unique
capacity of continuous 4D monitoring, thus bridging scientific research and operational applications. As for similar networks
in Europe and beyond, maturity of both instrumental technologies and physics-based data processing tools as the ones
described here suggest ALC/PLC networks could fruitfully contribute to aerosol measurements within the ACTRIS
European Research Infrastructure, at the same time representing a good example of earth observation science applications
for society.
**Code availability:** Codes used for data analysis can be provided upon request to the corresponding authors
**Data availability:** The presented datasets are made available to the Editor and Reviewers through a relevant shared directory
(credential provided). These will be made freely accessible and linked to a doi, should the revision process lead to a positive
outcome.
**Author Contribution:** Conceptualization, Data curation, Investigation: AnB, FB, HD**;** Formal analysis and Software: AnB;
Visualization: AnB, FB, HD**;** ALC instruments and database management: LDL, AlB, FP, HD, GPG; Funding acquisition
and Supervision: FB; Writing – original draft preparation: AnB, FB, HD; Writing – review & editing: AnB, FB, HD, AlB,
GPG.





**Competing interests:** The authors declare that they have no conflict of interest.
**Acknowledgements**
This research received partial financial support from the EC H2020 Project RI-URBANS (GA No 101036245), and
benefited from work done within the Action PROBE (CA18235), supported by COST (European Cooperation in Science and
Technology).
A. Bellini performed this work in the framework of her Doctoral Program at University 'La Sapienza', DIET, Rome, Italy.
A. Bracci and F. Pasqualini were supported by the project IR0000032 – ITINERIS, Italian Integrated Environmental
Research Infrastructures System, funded by EU - Next Generation EU, PNRR- Mission 4 "Education and Research" -
Component 2: "From research to business" - Investment 3.1: "Fund for the realisation of an integrated system of research
and innovation infrastructures".
We would like to thank: M. Clerico and D. Poggi (PLC-Torino), R. Cresta and A. Bisignano (PLC-Genova), E. Collino and
D. Perona (PLC-Milan-Rubattino), C. Cristofanelli (ALC-Mt.Cimone), S. Ottonelli (ALC-Taranto), C.R. Calidonna (ALC-
Lamezia Terme), M. Coltelli and R. Gueli (Catania and Etna ALC & PLC systems) for their contribution to the Alicenet
infrastructure, and L. Ferrero, A. Di Giosa, M. Furnari and G. Tranchida for support in the Alicenet sites of Milano Bicocca,
Rome Downtown, Messina and Capo Granitola, respectively.
PLC data in Milan-Rubattino, are collected by RSE in the framework of the 3-Year Research Plan 2022-2024 for the Italian
Electrical System (DM MITE n. 337, 15.09.2022), in compliance with the Decree of April 16th, 2018.
We acknowledge the Italian Air Force CAMM-Mt.Cimone for their support in the operation of the CMN-IT ceilometer,
funded by the Ministry of University and Research (MUR) by the Project "Potenziamento della Rete di Osservazione ICOS-
Italia nel Mediterraneo" PRO-ICOS_MED (PIR01_00019), and the GAW-WMO regional station "Rita Atria" for the
ceilometer hosting in Capo Granitola.
We also gratefully acknowledge S. Gilardoni, P. Bonasoni and A. Provenzale for providing the OPC data collected at the
'Testa Grigia' station at Plateau Rosa managed by the CNR Department of Earth System Sciences and Environmental
Technologies, and the ARPA Lazio for providing the Rome Tor Vergata TKE dataset.
We acknowledge the Copernicus Atmosphere Monitoring Service (CAMS) for the CAMS European air quality forecasts,
ENSEMBLE data: METEO FRANCE, Institut national de l'environnement industriel et des risques (Ineris), Aarhus
University, Norwegian Meteorological Institute (MET Norway), Jülich Institut für Energie- und Klimaforschung (IEK),
Institute of Environmental Protection – National Research Institute (IEP-NRI), Koninklijk Nederlands Meteorologisch
Instituut (KNMI), Nederlandse Organisatie voor toegepast-natuurwetenschappelijk onderzoek (TNO), Swedish
Meteorological and Hydrological Institute (SMHI), Finnish Meteorological Institute (FMI), Italian National Agency for New
Technologies, Energy and Sustainable Economic Development (ENEA) and Barcelona Supercomputing Center (BSC)
(2022): CAMS European air quality forecasts, ENSEMBLE data. Copernicus Atmosphere Monitoring Service (CAMS)
Atmosphere Data Store (ADS), https://atmosphere.copernicus.eu/.



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
