# Peer review of "Alicenet - An Italian network of Automated Lidar-Ceilometers for"

_EGUsphere, 2024_

## Author Comment (AC1)

Authors reply to: 'Comment on egusphere-2024-730', Anonymous Referee #2, 22 May 2024

Review of "Alicenet – An Italian network of Automated Lidar-Ceilometers for 4D aerosol monitoring: infrastructure, data processing, and applications" by A. Bellini et al.

This manuscript describes Alicenet, the network of Automated Lidar-Ceilometers (ALC) in Italy. The used instruments, sites, data quality control and processing are described, including the cloud screening, the denoising, the overlap correction and the absolute calibration. The retrievals of aerosol optical and physical properties are then presented with comparison with collocated or near-by Aeronet AOD and PM mass measurements. Alicenet also proposes an automatic identification of aerosol layers (ALADIN) with a distinction between the mixed, the continuous and the elevated aerosol layers. Case study and a 7 year climatology of the three layers are also described. The potential of Alicenet is finally emphasize by the description of the detection of a local dust front in the Po Valley, of a Saharan and biomass burning advection over Italy and of a volcanic eruption in Sicily. A lot of detailed information on the data processing, the retrieval and the automatic aerosol layers detection are given in the supplement.

The content of the manuscript suits AMT and the various tools developed by Alicenet are valuable. The manuscript is then worth publication in AMT.

We thank the reviewer for taking the time to revise our manuscript and for his/her constructive comments. Our point-to-point reply is given hereafter (in blue). The text lines and Figure references in our replies follow the updated numbering.

Major comments:

1. The general description of the network, the data processing, the aerosol retrievals and layers identification as well as a part of the technical descriptions are part of the manuscript whereas the detailed technical descriptions of the cloud screening, the overlap correction, the absolute calibration, the retrieval of aerosol properties and the identification of aerosol layers constitutes the supplement with additional figures. The supplement is then as important as the main paper for readers directly implicated in similar networks and related products. To some extent I really missed information of the supplement to practically understand e.g. some products during the reading of the manuscript. I would suggest to the authors to reconsider the subdivision of information between the main manuscript and the supplement. The criteria should be that the main manuscript allows to clearly understand the applied algorithms whereas only readers interested in reproducing the algorithms would refer to the supplement.

Thank you for this suggestion. We acknowledge that the subdivision between the main manuscript and the supplement was not optimal. In particular, for the algorithms description sections the submitted version had an hybrid approach, giving some details in the main text and some other in the supplement. Now, taking into account the comments by both Reviewers, we opted to restructure the manuscript so as to have a lighter main text in which the algorithms are only briefly introduced, moving all the more technical details in specific sections within the supplement. In this way, very expert readers as the Reviewers could find all the details in one place, possibly allowing them to reproduce the algorithms, while keeping the reading of the main text easier for the wider readership.

2. The English phrasing should be revised. Minor comments point to some (but not all) problematic sentences. Parenthesis are too extensively used. Acronyms are not always explained.

We revised the whole text and, based also on Rev.1 suggestion, decided to add a list of acronyms at the end of the text.

Minor comments:

Abstract:

L15: "in these years", english?

Thank you for this comment, we rephrased the sentence as (L15):
*'Since then, ALICENET grew up as a cooperative effort of Italian institutions dealing with atmospheric science and monitoring…'*

L24-26: The sentence "Examples of both …" could be merged with the previous one.

Done (L26):
*'…from near real-time monitoring to long-term analyses, examples of which are reported in this work'.*

L30: please remove a), b) and c).

Done

L32: suggest that???

Thank you for this comment, the sentence was not clear indeed. In the updated manuscript, this sentence was removed.

Introduction:

L38-39: two times "Particularly" in the same sentence.

Thank you, we rephrased the sentence (L39-41):
*Furthermore, high aerosol loads reduce visibility and, during major events such as desert dust storms, volcanic eruptions, and wide forest fires, can damage aircraft engines, thus representing a threat to the aviation sector (e.g. Flentje et al., 2010; Papagiannopoulos et al, 2020, Brenot et al., 2021; Monteiro et al., 2022, Ryder et al., 2024)*

L41: Consider deleting " represented by

We rephrased the sentence as (L41):
*'The vertical aerosol distribution is a key aspect to correctly quantify aerosol effects…'*

L 43-44: what is meant by "large-to-small scale atmospheric processes influencing local air quality" and by "it affects () high-elevation environments"?

Thank you for this comment, we rephrased the sentence (L42-45) to better explain the fact that the aerosol vertical distribution influences particle dispersion and transformation processes, such as the entrainment of lofted aerosol layers into the mixed layer, or particle formation and growth within stagnant layers (relevant references are reported in the text), and the state of high-altitude pristine environments, for example through aerosol depositions.

L 54 the acronym ACTRIS should be explained.

We explained the acronym in the text and added a list of acronyms at the end of the manuscript.

L55 ACTRIS or EARLINET ?

Thank you, we corrected the sentence (L57):
*'Infact, EARLINET lidar measurements are generally not performed continuously…'*

L57 variabilities?

Thank you, we rephrased the sentence (L58):
*'...to capture the high spatio-temporal variability characterising aerosols.'*

L64-65: I have the impression that the sentence missed the point that ALC's (originally conceived …) provide now complete aerosol backscatter profiles.

Thank you, we rephrased the sentence to better highlight the ALC potential (L63-65):
*'ALCs were originally conceived to only monitor the 'cloud ceiling', but recent technological improvements enabled ALCs to provide continuous quantitative information on aerosol properties profiles within the troposphere, including the boundary layer region…*

L75 remove such

Done

L78-81. Please rephrase

Thank you, we rephrased the sentence as (L77-80):

*'The development of such an extended ALC observational capacity was further accelerated after the eruption of the Icelandic volcano Eyjafjallajökull in 2010, which disrupted air transport due to the lack of readily accessible information on the horizontal and vertical displacement of the aerosol plume (Flentje et al., 2010, Mortier et al., 2013)'.*

L86 what is meant by "particles reaching down to be boundary layer"?

It was intended to describe the intrusion of aerosols from virgas into the boundary layer. Now corrected into (L84):
*'...particles reaching the boundary layer through evaporating rain…'*

L96-99: please rephrase

Thank you, we rephrased the sentence as (L95-97):
*'In Italy, an effort to coordinate ALC activities at national level and contribute to E-PROFILE has been done by the National Research Council - Institute of Atmospheric Sciences and Climate (CNR-ISAC), which set up the ALICENET network in 2015'.*

L112-115: please use a similar structure for sect2 to sect 5 or rephrase the whole §.

Thank you, the whole paragraph was rephrased as (109-114):
*'The work is organised as follows. Section 2 describes the ALICENET infrastructure. Section 3 introduces the main data processing steps and includes different examples of the relevant ALICENET products and accuracy. To facilitate the reading, the detailed technical aspects of each processing step were included in separated supplement sections (S1-S6), these being thus targeted to readers interested in a deep understanding of the processing chain, and possibly*

*in reproducing it. Sect. 4 shows three examples of the near-real time ALICENET monitoring capability, while Sect. 5 summarises the ALICENET achievements and some foreseen future developments within the network'.*

Alicenet sites and instruments:

Table 1: the beginning of measurements could be added since long-term time series are also discussed. If information from the supplement is moved to the manuscript, Table 1 could perhaps appear in the supplement in order to shorten the main paper.

Table 1 has been revised. It now also includes the beginning of measurements as usefully suggested. We believe this Table is important in the main text to give the general idea of the extension of the network, we therefore would prefer to keep it in the main text, also considering that the manuscript has been shortened as described in point 1.

Alicenet data processing and relevant products:

L163: are (aerosol + molecules) and (aerosol + molecular) both necessary ?

Thank you, we removed redundant details. The sentence now reads (L170-171):
*'Equation 1 allows to simply derive the total (i.e., aerosol + molecules) attenuated backscatter as follows…'*

L166: please rephrase: "Fig. 2 describes …" and remove "in this section".

Thank you, we rephrased L154-180 in the old manuscript (L158-182 in the new manuscript). The paragraph now starts with:
*'The ALICENET data processing chain is summarised in Fig. 2, with indication of main inputs and outputs.'*

L187: Batt: att in indices (please check the whole manuscript for exponent/indice).

Thank you, corrected in the whole manuscript.

Fig. 2: all used acronyms should be described in the figure or in the caption. E.g. PM, Sp, Vp, Mp, MAL, CAL, EAL,… are not described.

Thank you, acronyms descriptions have been added in all captions. Moreover, a list of acronyms was added at the end of the text.

L177-179: please rephrase

This part was moved and better explained in Sect. 3.4 (L500-503). It now reads:
*'Further discrimination of aerosol layers in terms of aerosol type could be derived exploiting PLC δv profiles. In fact, the inclusion of the PLC depolarisation information within the ALICENET processing is in progress, this representing a first step to automate the aerosol typing capacity within the network (thus complementing the aerosol layer typing capacity from more complex lidar systems, e.g., Nicolae et al., 2018; Córdoba-Jabonero et al., 2018)'.*

L179-180: this sentence could be deleted since it is explained in the figure caption.
Done.

L189: "and centralised at CNR-ISAC": already said.

Deleted.

L211 (if needed) has to be deleted

Deleted.

L214: give the definition of the SNR acronym the first time it is used

Thank you, the SNR is defined at L65 of the new manuscript. A list of acronyms was added at the end of the text to facilitate the reading.

L228-230: please rephrase

We rephrased the sentence as (L25-28 in supplement S2):
*'...windows in which a nearly homogeneous aerosol layer in the first 1200 m can be assumed. Then, based on this homogeneity assumption, for each selected time window it derives an overlap correction...'*

L244-246: change the sentence in order to remove the ()

We removed one parenthesis. The sentence now reads (L55-57 in supplement S2):
'Below 225 m a.g.l., the raw profiles are extrapolated down to the ground by linear fitting in winter (using data from 225 m to 285 m) or assuming a homogeneous profile below 225 m in summer, to avoid altitude ranges where the partial overlap is still insufficient to derive quantitative information.'

L258-260: Foehn events are then frequent and the aerosol load is then lower than in non-foehn event. Could you please add a reference for this statement.

Yes, Foehn winds coming from the Alps remove particles in the valley atmosphere (e.g., Mira-Salama et al., 2008). This reference was added to the  text.

L276-278: The word "reproduce" seems not correctly used. Alicenet correction produce a nearly-molecular correction with profile similar to the molecular profile but shifter to higher backscattering values due to the presence of weak aerosol load.

Thank you, we rephrased the sentence as (L258-261):
*'Overall, the results show that, while the manufacturer overlap function is unable to properly account for signal losses and leads to unphysical values lower than the molecular profile in the firsts 750 m, the $\beta_{att}$ profiles retrieved using the ALICENET overlap correction reasonably approach the nearly-homogeneous, nearly-molecular theoretical profiles expected in the selected episodes down to the ground.'*

L302-305: please summarize the QC2 control in the manuscript. It seems me important that the reader has a clear overview of the absolute calibration without referring to the supplement.

The basic principles of the absolute calibration procedure have been summarised in the main text to provide the reader with a clear overview (Sect. 3.2). As explained above, now all technical aspects of the algorithm implementation, including the detailed description of the calibration procedure and related quality controls have been moved to the supplement S3. A specific new Table was also introduced in the supplement (Table S6.1) to summarise all the QC of the Alicenet processing chain.

L312: the bar below Batt is strange and not described at L313

Thank you, corrected. (supplement S3, Eq. S3.3).

L316: supplement

Thank you, corrected.

Fig5: is it right that the CL value reported on Fig. 5a and 5b are the same ? this does not correspond to the text. Fig5c: Is there an explanation to the lower CL value at Roma and to the stronger seasonal cycle at Aosta and Messina?

Thank you for this comment. There was an error there. Figures 5a and 5b have now been corrected.

Also note that the instrument operating in Rome was manufactured well before the ones on Aosta and Messina, thus hardware changes might justify the lower CL values .

Concerning the seasonal cycle at Aosta and Messina, similar or even larger variability was also observed in other EU sites. The reasons for such a yearly cycle are currently under investigation in different European ALC networks. These aspects are now better explained in the main text with relevant references.

L329: A reference to PROBE documentation is needed.

Thank you, we included Buxmann (2024), Van Hove and Diémoz (2024) in the main text (L276-278).

L362, the "-1" should be in exponent.

Thank you, corrected (L124 in supplement S4)

3.3.1:

1) The impact of the iterative procedure to derive the aerosol backscatter and extinction has to be described. AOD with only constant LR could be also plotted on Fig. 6b and the results discussed. Is this impact similar for urban and remote stations (e.g. Roma and Aosta)? Are the constant LR method always worst than the iterative procedure (e.g. in presence of dust)?

Thank you for this comment.

It should be noticed that the impact of the vertically-variable LR in the retrieval of the columnar AOD may depend on the actual aerosol stratification. For example, if the atmospheric column is completely dominated by dust particles, the ALC-based AOD derived with the iterative procedure and a first guest LR of 38 sr may underestimate the reference sunphotometer AOD, while in the presence of inhomogeneous, non-dust aerosol profiles the ALICENET method has several advantages. However, we routinely apply this method because it is independent from ancillary data and a-priori assumptions, thus allowing an automatic, ALC-based retrieval of aerosol properties in different sites and periods.

We tried to answer the above questions in supplement S4.1 (L127-138) and further added a support Figure S4.1.1:

*'The iterative procedure has the ability to 'adjust' the first-guess, vertically-constant LR profile according to the actual aerosol stratification. An example of the 'adjusted', vertically-variable LR profiles is shown in Fig. S4.1.1a, this referring to the same period addressed in Fig. 6. It is worth mentioning that a main advantage of the variable-LR method is the fact that it is independent from ancillary (e.g., sunphotometer) data and a-priori assumptions (e.g., the actual LR value to be used), thus allowing an automatic, ALC-based, homogeneous retrieval of aerosol properties in different sites and periods. As reported in the main text (Fig. 7) and further shown in Dionisi et al. (2018), the*

*ALICENET-retrieved aerosol optical properties were found in good agreement with independent sunphotometer data in different sites. A comparison of the performances of the adjusted-LR and fixed-LR approaches was also conducted by Dionisi et al. (2018) using ALC data from different stations. In brief, the authors found a good agreement between sunphotometer and ALC-based AOD using both the iterative procedure and a fixed LR of 38 sr, and larger discrepancies using a fixed LR of 52 sr, regardless of the site location. As an example, in Fig. S4.1.1b we show a short-term comparison of the AOD retrieved with both ALICENET processing and fixed-LR values (chosen as in Dionisi et al., 2018) and the reference AERONET L2 AOD in the same period presented in Fig. 6.'*

2) Fig 6b shows cases where AOD from Aeronet is not comprised within the expected uncertainties of ALC AOD (L377-378). There is an underestimation in case of dust that is described in the manuscript. What about the overestimation systematically found at around midday (Fig. 6b), particularly on the 2017-07-13?

Thank you for this comment. In general, possible reasons for the ALC overestimations are: a) ALC signal noise around midday, and b) different portions of the atmosphere sounded by the vertically-pointing ALC and the sunphotometer.

It should be noticed that AERONET L2 data are not available when the ALC derives an AOD increase on 2017-07-13 afternoon (to facilitate the reading, Fig. 6b has been improved).

Moreover, as reported in the caption of Fig. 6b, error bars refer to the AOD standard deviations within the 1 hour averaging interval. The AOD uncertainty conservatively declared by ALICENET is 30-40%, thus larger than the error bars plotted in Fig. 6b.

3) Fig. S3 shows that Roma and Messina have much more coarse mode aerosol and both stations have a better correlation between AOD from Alicenet and Skynet. This should be the inverse since continental aerosol properties were chosen for the retrieval. Could you please comment on that point?

Thank you for this comment. In the main text (L347-348) we tried to answer the above question (L345-347):

*'..underestimations are mainly related to the presence of non continental aerosol types, such as dust and marine particles in Messina, or shallow aerosol layers in the blind overlap region (i.e., below 225 m a.g.l.), as is the case of Aosta during winter (see Fig. 9).'*

Figure 9 shows the comparison between the ALC and OPC mass estimates at the ground level in Aosta: main ALC underestimations are found in January-February, November-December.

Moreover, we quantified the percentage of data pairs lying within $\pm 0.01 \pm 0.15*AOD_{sunphotometer}$ from the 1:1 line (L342). The percentage is higher in Aosta (84%) with respect to Rome and Messina (73% and 70%, respectively).

L390: S4

Thank you, corrected.

L409: Consider replacing "when the two are to be compared" by something similar to "before comparing them" or "when comparing them".

Thank you. Replaced with 'before comparing them'.

L428: to which type of aerosol corresponds the chosen density? Please also indicated the aerosol-type of the chosen gamma exponent (L431) and what is MERIDA.

In supplement S4.2 (L171-175) we included more details:

*'In that case, γ = 0.2 in the presence of continental, hygroscopic aerosols (D'Angelo et al., 2016) and γ = 0 (i.e., Mpdry = Mp) in the presence of dust, hydrophobic particles (Barnaba et al., 2010). The aerosol type was assessed through the linear volume depolarisation ratios (δv) profiles of a co-located PLC, assuming that aerosol mixtures associated with δv < (>) 15% are dominated by secondary (dust) particles. The RH was extracted from the dataset of the high-resolution atmospheric model MERIDA (Bonanno et al., 2019).'*

Moreover, to be coherent with the above approach, we updated the results shown in Fig 8 using different aerosol densities for dust and non-dust particles (main text, L371-372):

*'The aerosol density used to derive both ALC and OPC aerosol mass concentrations was 1.2 (1.6) g cm-3 in the presence of non-dust (dust-dominated) aerosol mixtures (Diémoz et al., 2019b).'*

L433-435: please rephrase and consider using a relative sentence.

We rephrased the sentence as (L375-379):

*Fig. 8 shows that the two mass concentration series exhibit similar time evolution, with good agreement both in low aerosol conditions (e.g. 6-15 June 2022), and during transport events increasing the local aerosol load. In the considered period, main transport events were associated with desert-dust intrusions (e.g., 3-5, 18-22, and 27-28 June 2022) and Po Valley pollution advections (e.g., 13-14, and 25-26 June 2022).*

Fig. 8 and related description: it would be nice to have an estimation of the maximal difference between ALC and OPC mass estimates and of these differences as a function of humidity in order to estimate the adequation of the humidity correction

We tried to answer the above questions in supplement S4.2 (L175-180) and further added a support Figure S4.2.2:

*'During the period addressed in Fig. 8, at the altitude of Testa Grigia - Plateau Rosa the simulated RH ranged from 16% to 98%, and the measured δv from 0.4% to 27%. In Fig. S4.2.2a we further show the same data including both RH-non corrected (wet, blue bullets) and RH-corrected (dry, red bullets) aerosol mass concentrations as retrieved by ALICENET. The median difference between the ALC-based wet/dry aerosol mass concentrations and the OPC PM10 measurements was evaluated as a function of RH in Fig. S4.2.2b. It shows the median differences between ALC- and OPC-based aerosol mass concentrations per RH bins, with dry values keeping around zero. On average, the hygroscopic correction reduced the difference between ALC and OPC mass estimates, but the large horizontal distance between Aosta and Testa Grigia - Plateau Rosa and the uncertainty of the ALICENET aerosol volume retrieval in dust conditions strongly complicate the evaluation of this correction.'*

.

-L459-461: please rephrase

See below response.

-L459-475: please consider using the name of the source of uncertainties instead of the first, second, …factor.

Thank you for the above comments, the whole Sect. 3.3.3  was rephrased as:

*'The previous sections describe the ALICENET efforts to exploit the great potential of ALC in providing quantitative aerosol-related geophysical parameters, and demonstrate the good performances of the current algorithms. Nonetheless, due to several factors also discussed above, the expected uncertainties associated with the output products range from 20% for the attenuated backscatter (product L2 in Fig.1) to 50% for the aerosol mass (L3 in Fig. 1). The main factors are listed hereafter.*

*1) the instrumental noise of the signal. This factor depends on the instrument status and mainly impacts the retrievals  in the middle-upper troposphere.*

*2) the overlap correction applied to the signal. As discussed, this factor is critical in the lowermost levels and accurate instrument-specific, overlap-correction models are necessary to derive quantitative information in the first 800 m. Accuracy of the retrievals in this vertical region depends on the statistical and physical representativeness of the ensemble of overlap functions from which the overlap model is derived (supplement S2).*

*3) the variability of the instrument calibration coefficient. This third factor (see Sect. 3.2), directly impacts the accuracy of $\beta_{att}$. For example, it is found by error propagation that changes of 30% in the instrument calibration coefficient (which are quite usual in some ALICENET and E-PROFILE stations) translates into a variability in  $\beta_{att}$ up to 20%.*

*4) the accuracy of the functional relationships used in ALICENET to link the aerosol backscatter to the other aerosol properties, impacting the estimation of $\alpha_p$, $S_p$, $V_p$, $M_p$ and, to a lesser extent, $\beta_p$. This factor strongly depends on the actual aerosol conditions: the functional relationships can give a good estimate of the aerosol properties in presence of continental aerosols, while in presence of non-continental particles they are less accurate (a relative error of 30-40% was derived by Dionisi et al., 2018). As mentioned, extension of the ALICENET approach to include other aerosol types is foreseen for the next future. In particular, exploitation of the PLC depolarisation profiles for aerosol-typing will drive the selection of aerosol-type specific functional relationships (e.g. Gobbi et al., 2002).*

*Concerning the retrieval of aerosol mass concentrations, the assumed particle densities are a major source of uncertainty, and the accuracy of the retrieval depends on the possibility to better constrain the aerosol density profiles, e.g., through ancillary data, including depolarisation information.*

*Overall, the above factors result in instrument-, time- and range-dependent uncertainties of the ALC-based aerosol optical and physical properties. The expected uncertainty with an optimal SNR up to at least 7 km a.g.l., an overlap error < 10% in the lowermost levels, and in presence of continental aerosol types is of 20% for $\beta_{att}$, 30-40% for AOD, reaching 50% for aerosol mass.'*

L476-478: what about the uncertainty of the humidity correction?

Thank you for this comment. In supplement S4.2 (L162-165) we included:

*'The values of the dry aerosol density $\rho_d$ and of the $\gamma$ exponent depend on the aerosol mixture under investigation and are the main sources of uncertainty in the aerosol mass retrieval (Adam et al., 2012; see also main text, Sect. 3.3.3). Their accuracy strongly depends on the possibility to identify the actual, dominant aerosol type, e.g., through depolarisation information and/or model data'.*

We decided to keep all information on the hygroscopic correction in the supplement S4.2, since this correction is not performed routinely within ALICENET processing and rely on the availability of ancillary data (e.g., RH values). In the reference paper Adam et al. (2012) the authors reported an uncertainty of the hygroscopic correction for aerosol optical properties between 30%-40%.

-L485-487: please rephrase

Thank you, we rephrased the sentence as (L443-447 in main text):

*'As already mentioned, a main advantage of ALCs is their ability to operate continuously, which allows detecting and tracking the variability of the aerosol vertical stratifications at multiple timescales using aerosol as passive tracers. This information can be beneficial for several sectors, among which AQ and meteorology (e.g., Moreira et al., 2019; Ravnik et al., 2024; Körmöndi et al., 2024), aviation (e.g., Osborne et al., 2019; Salgueiro et al., 2023), atmospheric research (e.g., Jozef et al., 2024)'.*

L 490-508 The expression mixed layer or mixing layer is more often used than mixed boundary layer. Generally, the use of standard names and acronyms would be helpful (see e.g. Kotthaus et al., 2023 that you cite), e.g. MAL is rarely used. It should also be worth relating your atmospheric layers to standard ones: is MAL similar to your previously mentioned MBL and to ML or convective boundary layer? If not (L517-518), what is the link/interaction between them? Which conventional layers are comprised into the CAL?

In the updated Sect. 3.4, we tried to better explain why we prefer to use a classification (and related naming) based on aerosols (e.g., MAL, CAL) rather than a more conventional one based on thermodynamics (e.g., ABL, ML). We also tried to explain the link/interaction between them (lines 448-482):

*'Commonly identified  atmospheric stratifications based on ALC data analysis include the Atmospheric Boundary Layer and the Mixed Layer (ABL and ML, respectively, e.g., Poltera et al., 2017; Kotthaus et al., 2020; Caicedo et al., 2020), and lofted aerosol layers in the free troposphere (e.g., Adam et al., 2020). The ABL is a thermodynamic layer connected to the Earth's surface and capped by a temperature inversion, while the ML is an ABL sublayer mixed by turbulent fluxes (Stull, 1988; Kotthaus et al., 2023).*

*However, it should be noticed that aerosols are 'delayed' tracers of atmospheric dispersion processes and may not always consistently represent the thermodynamic state of the atmosphere (Haeffelin et al., 2012). The tracking of thermodynamic layers through aerosol lidars can be complicated by superimposing phenomena such as large-to-medium scale advections, natural and anthropogenic emissions, particle physico-chemical transformations. These processes may remove or transport particles in specific atmospheric ranges (e.g., Collaud Coen et al., 2018; Diémoz et al., 2019a), modulate the daily cycle of aerosol profiles (e.g., Diémoz et al., 2021), form aerosol layers within and above the ABL (e.g., Curci et al., 2015; Sandrini et al., 2015), thus decoupling the aerosol-related and thermodynamic stratifications.*

*This decoupling is expected to be further enhanced over complex terrain (e.g., Serafin et al., 2018) and/or over regions affected by multiple natural and anthropogenic sources, as is the case of the Italian territory.*

*For all these reasons, the choice for aerosol layers detection and naming in ALICENET was to keep a clear link to the aerosol field allowing its identification, avoiding a terminology traditionally based on thermodynamics. In particular, we develop a novel Aerosol LAyer DetectIoN (ALADIN) tool to automatically derive aerosol layering information from ALCs/PLCs across the network, this targeting the following aerosol layers:*

1. *the Continuous Aerosol Layer (CAL): it is the layer extending from the ground level and characterised by the continuous presence of aerosols along the vertical profile;*
2. *the Mixed Aerosol Layer (MAL): it is a CAL sublayer within which aerosols are mixed by surface-driven turbulent fluxes;*
3. *Elevated Aerosol Layers (EALs): they are lofted aerosol layers which lie above the MAL, and either within or above the CAL.'*

For the reasons reported above, the link between CAL/MAL and standard thermodynamic layers (e.g., ABL/ML) is not always obvious. In general, the MAL and ML exhibit a similar daily cycle, and the CAL can sometimes be linked to the ABL, although this result can not be generalised.

L533: Does "centre" mean "center"? If yes, please use the English word. Otherwise, please explain the significance or acronym.

Thank you. We used British vocabulary in the revised version.

Moreover, in the updated manuscript the expression 'layer center' was removed since it was redundant.

L534-546: this § is quite difficult to follow. I understand that specific details and detailed algorithms are only described in the supplement. The data treatment in Alicenet and particularly in ALADIN should however remained comprehensive in the main manuscript.

A schematic description of ALADIN can perhaps be helpful. I suppose that some restrictions like MAL<CAL<EAL are also applied? L553-555 and Fig. 10 allows to see than EAL<CAL. In usual ABLH description, the residual layer is not considered as an elevated aerosol layer. Only layers above the CBL and RL are described as EAL (see remark on L490-508).

As mentioned at the beginning, we are aware that the original structure made the technical details of the algorithms, including ALADIN, difficult to follow. Now all data treatment within ALADIN has been moved to supplement S5. Following the reviewer suggestion, we also added there a schematic description of the ALADIN processing (Fig S5.2).

The restrictions applied for the search of the CAL, MAL, and EAL layers are reported in supplement S5. In brief, it must be MAL<CAL, and EAL>MAL. Hence, EALs can also be detected below CAL, for example in the case of low-level advections or particle formation and growth within the CAL (see, e.g., Fig. 10).

Equ. 6 is the median done only on the numerator? If the median of the whole expression is taken, please use median (…) instead of the misleading overline.

Corrected using median(...)

L571: CAL diurnal cycle is clearly visible only in June, July and August. The thicker CAL during night in winter is not obvious.

This behaviour can be attributed to particle hygroscopic growth during nighttime. Other possible reasons are currently under investigation

L580-582: Even if there is an increase of aerosol emission at ground in the early morning, what is the mixing mechanism allowing aerosol to reach a higher altitude than TKE during the first part of the day?

In addition to the turbulent mixing of aerosol particles, possible aerosol-related mechanisms favouring the MAL growth during the morning are secondary aerosol formation (e.g., Nilsson et al., 2000) and hygroscopic growth (e.g., Curci et al., 2015). These mechanisms lead to an increase of the available aerosol particles which can be dispersed by turbulent fluxes.

Please note that the MAL and CAL heights reported in Figure 10 and 11 of the revised manuscript are slightly different with respect to the ones reported in the first manuscript. Infact, in the last months we reprocessed and quality-controlled the multi-annual dataset of different ALICENET systems for the long-term analysis of aerosol profiles (work in preparation). The data used in Rome come from the reprocessed dataset.

Concerning the land-sea breeze, the afternoon should lead to a breeze from sea to land during the afternoon (=sea-land breeze?). Could the land to sea breeze in the morning explain the measured effect?

Yes, a sea-land breeze regularly develops in the considered site during the afternoon.

The land-sea breeze during the morning generally interests a thin atmospheric layer near the surface. Its effects on the CAL/MAL heights is currently under investigation.

Fig. 12b: the mention of the station (Milan) and of the time on the webcam pictures will be helpful.

Done, thank you.

L749 ", ,"

Corrected.

Summary and future perspective:

The future perspectives are well described. A more extended mention of the data chain, of the aerosol optical and physical retrievals and of ALADIN is however necessary for reader restricting themselves to the summary in order to deserve Alicenet's potential.

We thank the referee for this suggestion, aimed at the valorisation of the ALICENET outputs in the conclusion. We think that the following sentence in the last section, which was slightly revised, could fit the purpose (L649-662):

'*Since the beginning of the ALICENET activities, particular care has been devoted to data retrievals and exploitation, this also taking advantage of technical/scientific exchanges within European initiatives, such as the EC Cost Actions TOPROF (2013-2016) and PROBE (2019-2024), the ongoing EUMETNET program E-PROFILE (2020-2028) and the EC H2020 Project RI-URBANS (2021-2025). In this context, ALICENET developed a specific, centralised and automated data processing chain with associated data quality control (QC) procedures, as presented in detail in this*

*work. The data processing steps were either refined from previously published work (e.g. Hervo et al., 2016, Dionisi et al., 2018), or are completely new, as the automatic aerosol layers detection algorithm (ALADIN). Overall, the processing chain includes signal correction and calibration procedures (Sects. 3.1, 3.2), the aerosol properties inversion (Sect. 3.3), and the identification of vertical stratifications (Mixed, Continuous and Elevated Aerosol Layers, MAL, CAL and EALs, respectively, Sect. 3.4). Output products with different levels of complexity and associated uncertainties are thus provided (Fig. 2). These range from more basic L1 quantities (as the Range-Corrected Signal, RCS, and, where applicable, depolarisation, $\delta_v$), through the L2 total attenuated backscatter $\beta_{att}$ to the L3 aerosol optical ($\beta_p$, $\alpha_p$ and thus AOD) and physical ($S_p$, $V_p$, and $M_p$) properties plus vertical layering'.*

Supplement:

(1): to be coherent with the text, the c of Fc(r) should be written as indices.

Corrected.

L 26: over an ALC dataset longer than one year.

Corrected.

L29-36: The description of the applied overlap correction is quite difficult to follow. Please reformulate so that the same procedure can really be applied without potential error. Your condition of filtering out the overlap corrections associated with Mov < 0.05 means that you select RD(r) sufficiently different than RDnear(r) or that the T difference with the internal T is sufficiently large?

We reformulated the sentence and the equation (L36-43):

*'..., for each $f_c(r)$ the following metric is calculated:*

$$M_{ov} = \sum \left( \frac{median(|RD(r) - RD_i(r)|)}{|T_{instr} - T_i|} \right)/N \qquad \text{(Eq.S2.2)}$$

*where RD and Tinstr are the relative difference and the system internal temperature associated with the considered fc(r), the median is calculated over the vertical range 225-1200 m, and the sum is performed over the sub-ensemble of system internal temperatures Ti and associated relative differences RDi lying between ± 5 K from the considered Tinstr. Then, fc(r) associated with Mov < 0.05 are rejected'*

This condition allows us to filter out the outliers from the ensemble of overlap functions. The outliers are the RD(r) which deviate from the other members of the ensemble having similar Tint.

L45-59: too long sentence

We reformulated and split the sentence (L67-72):
*'Along this vertical range, an iterative procedure is applied over an ensemble of 'potential' molecular windows centred at different altitudes and with variable amplitudes, these ranging from 600 to 3000 m at steps of 30 m. For each potential range-window, i.e., combination of central altitude and amplitude, the linear fit between the time-window-averaged signal and the theoretical molecular attenuated backscatter profile is performed. In order to reject those range-windows still affected by aerosol loads, a test is performed to check for the presence of coherent structures therein'.*

L49: "is performed"

*Rephrased (L72-73):*
*'More specifically, the Breusch-Godfrey test (BG test; Breusch, 1978) is applied to calculate the autocorrelation in fit residuals'.*

L50: please add the name and the formula of the adjusted R2 parameter so that a similar procedure could be applied without using the bgtest R package.

The formula has been added (supplement S3, Eq. S3.2)

L55: To this purpose ??

Corrected (L72-73).

L54: A mention of QC1 could be helpful

All QCs applied within the calibration procedure are now described in supplement S3, and further summarised in Table S6.1 in the supplement.

(4): please put "slope" as exponent.

Corrected (Eq. S3.4)

L62: if CL is a median, please provide equ.(4) including the median.

Thank you, the median was included in the equation (Eq. S.3.4) and in the text.

S3: please add AOD in the x and y labels

L90: why is CAL mentioned but not described? If described in the manuscript, please add a sentence with a reference to the CAL description.

Thank you, all procedures used within ALADIN are now described in supplement S5, including the CAL detection

L108: SD in indices?

Corrected. The symbol is now $\sigma_w$ (introduced at L213 in supplement S5)

L110-111 should probably be aligned similarly to L108-109

Corrected

L112: are you sure that an additional continuity criterion will be benefitable?

Thank you, the sentence was removed since, as observed by the reviewer, it was not supported by scientific evidence.

L122: if 'centres' means centers, please change the world into the English's one.

We use British vocabulary in the revised version. In the current version of the supplement, the expression 'layer center' was removed since it was redundant.

L126 : the potential bottom and top semi-amplitudes of 150-1500 m means that the EAL thickness is estimated

between 300-3000 m? or that its elevation has to be comprised between 150-1500 m? Please clarify.

Thank you, the sentence was not clear. We mean that the EAL thickness is estimated between 300-3000 m. In the updated supplement S5 the description of the ALADIN EAL detection was revised (lines 234-258) to better explain the main steps of the procedure. The values of the maximum and minimum bottom/top semi-amplitudes were removed since they are user-defined.

7. Why not using the usual integral symbol?

The integral symbol has been used in the updated supplement (Eq. S5.2)

L131-136: Will it be easier for the reader to give the equation instead of a description?

The sentence has been rephrased, also using mathematical notation.

---

## Author Comment (AC2)

Authors reply to 'Comment on egusphere-2024-730', Anonymous Referee #1, 23 May 2024

The manuscript describes the Italian network of automated lidar and ceilometer measurements, Alicenet. The network consists of state-of-the-art ceilometers distributed from the north to the south of the country. The manuscript describes the quality control of the data and the algorithms to derive optical and physical aerosol properties, as well as a new procedure for aerosol layer identification. The derived properties are compared to AERONET AOD and in-situ measurements for validation. Long-term monitoring over seven years and several case studies show the capabilities of the network.

The algorithms used in the manuscript are based on established methods used in other lidar and ceilometer networks, but were further developed to improve the results derived from the relatively simple ceilometer instruments having only one wavelength. These algorithms have the potential to be included to the wider e-profile network and to provide further products on a European scale.

The manuscript is suitable for publication in AMT, but may be considered to be divided into a more technical and an experimental part.

We thank the reviewer for taking the time to revise our manuscript and for his/herheir constructive comments. Our point-to-point reply is given hereafter (in blue). The text lines and Figure references in our replies follow the updated numbering.

Major comments:

Did all instruments measure at all times? Or were some added later? The specific measurement periods or the start of the measurements could be added to Table 1.

Table 1 has been revised. It now also includes the beginning of measurements as usefully suggested. We believe this Table is important in the main text to give the general idea of the extension of the network, we therefore would prefer to keep it in the main text, also considering that this has been shortened as described below.

The description of the technical details of the data quality control and the algorithm development may be moved to the supplement to improve the readability of the manuscript, as already suggested by another review.

Thank you for this suggestion. We acknowledge that the subdivision between the main manuscript and the supplement was not optimal. In particular, for the algorithms description sections the submitted version had an hybrid approach, giving some details in the main text and some other in the supplement. Now, taking into account the comments by both Reviewers, we opted to restructure the manuscript so as to have a lighter main text in which the algorithms are only briefly introduced, moving all the more technical details in specific sections within the supplement. In this way, very expert readers as the Reviewers could find all the details in one place, possibly allowing them to reproduce the algorithms, while keeping the reading of the main text easier for the wider readership.

Throughout the whole manuscript: the language should be improved, sometimes this makes the understanding of the meaning difficult. Too many parentheses, as already stated in the quick review.

Thank you for this suggestion. We revised the manuscript to improve language and phrasing.

Minor comments and some suggestion:

Line 13: these -> such systems

Thank you, the sentence now reads (L12-13):

*'This information is primarily derived by lidar active remote sensing, in particular with extensive networks currently in operation worldwide'.*

Line 17-20: long sentence

Thank you. The sentence has been rephrased (L17-20):

*'In the current configuration, the network makes use of both single-channel ALCs and dual channel, polarisation-sensitive systems ALCs (referred to as PLCs). The systems operate in very different environments (urban, coastal, mountainous and volcanic areas) from Northern to Southern Italy, thus allowing the continuous monitoring of the aerosol vertical distribution across the country'*

Line 21-24: long sentence

The sentence has been rephrased (L23-25):

*'In this work, we present the ALICENET infrastructure and the specifically-developed data processing centralised at CNR-ISAC, converting raw instrumental data into quantitative, quality controlled information on aerosol properties, ranging from attenuated backscatter to aerosol mass and vertical stratifications'*

Line 245: Overall, this setup provides from near real-time to long-term overviews of the 4D aerosol field over Italy. ...

is difficult to follow.

Maybe rather:Overall, this setup provides near real-time as well as to long-term overviews of the 4D aerosol field over Italy.

Thank you. The sentence has been rephrased (L23-25):

*'This setup allows to get insights into the 4D aerosol field over Italy with applications from near real-time monitoring to long-term analyses, examples of which are reported in this work.'*

Line 27-30: long sentence

Thank you. We tried to slightly abbreviate the sentence (L29-33):

*'Overall, ALICENET represents a valuable resource to extend the current aerosol observational capabilities in Italy and in the Central Mediterranean area, and contributes to bridge a gap between atmospheric science and its application to specific sectors, among which air quality, solar energy, aviation safety.'*

Line 31: of the data processing tools available -> of the available data processing tools

The sentence at L30-33 was removed.

Line 32: "could usefully integrate", what do you mean? ALC networks can integrate other EU infrastructures, or vice versa?

We removed this sentence, since it was not clear.

Some specific remarks:

Line 132: "having a sufficiently high SNR" what is sufficient? At what altitudes? Any reference in the literature?

Thank you for this comment. We intended systems that allow to probe at least up to the middle troposphere, also for calibration purposes (e.g., Wiegner et al., 2014). The sentence was modified in order to clarify this aspect (L131-133):

'...it was agreed to operate standardised systems across the network choosing the ones that allow to probe at least up to the middle troposphere, also for calibration purposes (e.g., Wiegner et al., 2014; see also Sect. 3.2)'.

Table 1 and, among others, L 135: Change CHM15K to CHM15k

Done.

Eq. (3) and line 313, where is the overbar? Below beta?

Thank you, Eq. (3), now Eq. (S3.2) in supplement S3, was updated using median(...).

Fig. 5: Are the CL values really the same? Minimum and maximum for spring and autumn, as stated in the text? I would recommend to add date and time information on top of each subplot a) and b) for easier visual identification.

Thank you for this comment. There was an error there. Figures 5a and 5b have been corrected and date and time information added.

Line 358: More specifically, an iterative procedure is used to derive $\beta$p(r) and $\alpha$p(r) vertical profiles.

Corrected, the sentence now reads (L119-120 in supplement S4.1):

*'Operatively, an iterative procedure is applied within the forward Klett inversion to derive βp(r) and αp(r) vertical profiles as follows…'*

Line 391: correct notation would be: Ångstrøm

Thank you, corrected.

Section 3.4. the use of "new" names and abbreviations for the different boundary layer parts is misleading. I would suggest to use the established ones as: planetary boundary layer - PBL, mixing layer - ML, residual layer, to avoid confusion. CAL, EAL, MAL are not very common.

We acknowledge that the classification of layers based on a thermodynamic description of the atmosphere (PBL, ML, residual layer, etc.) is more established. However, as noted by other authors (e.g., Haeffelin et al., 2012), aerosols are not perfect proxies for atmospheric thermodynamics, and their concentrations can sometimes be decoupled from atmospheric dynamics. Therefore, we chose to retain the current terminology based on direct observations by the ALCs, which we find more appropriate. We have revised the text in Sect. 3.4 to clarify this choice for the reader.

Line 549 and Eq. 6: again: overbar for median and division lines are not clear.

Thank you, in the updated version we used median(...) (see Eq. S5.3 in supplement S5).

Line 749, two , ,

Thank you, corrected.

Line 705: Italian observational gap at EU level. This has been emphasised 3 times. Was there a gap before, and it is now filled since 2017?

Thank you for this comment. It was intended to emphasise the fact that in Italy the monitoring of aerosol vertical profiles is not yet coordinated by a national weather service. An effort to coordinate these activities, collect data at national level, and cooperate with the EU community was conducted by ALICENET.

---

## Author Response (AR1)

Dear Editor,

We fully revised our manuscript entitled 'ALICENET - An Italian network of Automated Lidar-Ceilometers for 4D aerosol monitoring: infrastructure, data processing, and applications' (EGUSPHERE-2024-730) following the two Reviewers suggestions.

Both of them raised the point of difficult reading of the ALICENET processing algorithms descriptions in relation to the chosen separation between main text and supplement . However, while one Referee seems to suggest to provide more details in the main text ('To some extent I really missed information of the supplement to practically understand e.g. some products during the reading of the manuscript'), the other suggested the opposite ('The description of the technical details of the data quality control and the algorithm development may be moved to the supplement to improve the readability of the manuscript, as already suggested by another review').

We acknowledge that the subdivision between the main text and the supplement was not optimal in the submitted version of the manuscript. In fact, the algorithms description sections had a confusing hybrid approach, giving some details in the main text and some others in the supplement.

Taking into account these useful comments of both Reviewers, in this revised version we thus opted to extensively restructure the manuscript. It now has a lighter main text in which the algorithms are only briefly introduced, and all the technical aspects of the ALICENET algorithms were moved into detailed, specific supplement sections. We believe that this makes the reading of the main text easier for the wider readership interested in getting information about ALICENET, its products and related applications. At the same time, very expert readers (as the Reviewers) could find all the specific technical details in one place (the supplement), this potentially allowing them to reproduce the algorithms.

A detailed list of modifications is given in the point-by-point reply to Referees.

Due to the important revisions of both the structure and language of the main manuscript and the supplement, at this stage we are only submitting the 'clean' version of revised manuscript, as the additional 'track-changes' version will likely generate confusion.

Finally, we would like to point out that, as stated since the submission phase, it is our intention to make the presented datasets  freely accessible and linked to a doi, should the revision process lead to a positive outcome. At the moment, these are still accessible to the Editorial Office and to the Reviewers at the link provided at the submission.

Thank you for your support in the revision process

Annachiara Bellini and Francesca Barnaba on behalf of all co-authors